# Evaluation of the clinical utility of the PromarkerD in-vitro test in predicting diabetic kidney disease and rapid renal decline through a conjoint analysis

**Lauren Fusfeld**[1]*, **Jessica T. Murphy**[1], **YooJin Yoon**[1], **Li Ying Kam**[2], **Kirsten E. Peters**[2], **Pearl Lin Tan**[2], **Michael Shanik**[3,4], **Alexander Turchin**[5]

1 Boston Healthcare Associates (now a Veranex Company), Boston, MA, United States of Ameria, 2 Proteomics International, Perth, Australia, 3 Stony Brook University Medical Center, Stony Brook, NY, United States of Ameria, 4 Endocrine Associates of Long Island, PC, Smithtown, NY, United States of Ameria, 5 Brigham and Women's Hospital, Boston, MA, United States of America

* lfusfeld@bostonhealthcare.com

**Data Availability Statement:** All relevant data are within the manuscript and its Supporting Information files.

## Abstract

### Background

Early identification of patients at risk of developing diabetic kidney disease or rapid renal decline is imperative for appropriate patient management, but traditional methods of predicting renal decline are limited.

### Objective

This study evaluated the impact of PromarkerD, a biomarker-based blood test predicting the risk of diabetic kidney disease (DKD) and rapid renal decline.

### Methods

Conjoint analysis clarified the importance of PromarkerD and other patient attributes to physician decisions for type 2 diabetes patients. Forty-two patient profiles were generated, with varying levels of albuminuria, estimated glomerular filtration rate (eGFR), blood pressure, hemoglobin A1c (HbA1c), age, and PromarkerD result. A web-based survey asked each physician to make monitoring/treatment decisions about eight randomly selected profiles. Data were analyzed using multivariable logit models.

### Results

Two hundred three primary care physicians and 197 endocrinologists completed the survey. PromarkerD result was most important for increasing the frequency of risk factor monitoring. PromarkerD was second to HbA1c in importance for deciding to prescribe sodium/glucose cotransporter-2 inhibitors (SGLT2s) with a DKD indication, second to blood pressure for increasing the dose of lisinopril, and second to eGFR for replacing ibuprofen with a non-nephrotoxic medication. Compared with no PromarkerD results, a high-risk PromarkerD

**Funding:** Funding for this study was provided by Proteomics International (URL: https://www.proteomics.com.au/) to Boston Healthcare Associates (BHA). Proteomics employs LYK, KEP, and PLT, who had a role in influencing study design, the decision to publish, and the preparation of the manuscript. The funders had no role in data collection and analysis.

**Competing interests:** LF and JM are employed by BHA, and YY was employed by BHA at the time of the research. Proteomics employs LYK, KP, and PLT. Dr. Shanik has received consulting fees from Proteomics International. Dr. Turchin has received consulting fees from Proteomics International, has equity in Brio Systems, and has received research funding from Astra Zeneca, Edwards, Eli Lilly, Novo Nordisk, Pfizer, and Sanofi. Proteomics International is a beneficiary of patent PCT/AU2011/001212 that relates to biomarkers described in this manuscript. This does not alter our adherence to PLOS ONE policies on sharing data and materials.

result was associated with significantly higher odds of increasing monitoring frequency (odds ratio [OR]: 2.56, 95% confidence interval: 1.90–3.45), prescribing SGLT2s (OR: 1.98 [1.56–2.52]), increasing lisinopril dose (OR: 1.48 [1.17–1.87]), and replacing ibuprofen (OR: 1.78 [1.32–2.40]). A low-risk PromarkerD result was associated with significantly lower odds of increasing monitoring frequency (OR: 0.48 [0.37–0.64]), prescribing SGLT2s (OR: 0.70 [0.56–0.88]), and replacing ibuprofen (OR: 0.75 [0.57–0.99]).

## Conclusion

PromarkerD could increase adoption of renoprotective interventions in patients at high risk for renal decline and lower the likelihood of aggressive treatment in those at low risk. Further studies are needed to assess patient outcomes with PromarkerD in real-world practice.

## Introduction

Type 2 diabetes mellitus (T2DM) is a major health problem; approximately 33 to 35 million Americans have type 2 diabetes mellitus [1]. One of the main complications for type 2 diabetes mellitus is diabetic kidney disease, a chronic and progressive kidney disease associated with both type 1 and type 2 diabetes mellitus. Diabetic kidney disease has become a public health concern as the number of adults with type 2 diabetes mellitus has grown over the past twenty years [2, 3]. Currently, 40% of people with type 2 diabetes mellitus have diabetic kidney disease, and this prevalence is projected to grow to 50% by 2025 [4]. Diabetic kidney disease is the leading cause of end-stage renal disease (ESRD) and associated with increased morbidity and mortality [3, 5]. Furthermore, diabetic kidney disease and ESRD impose economic burdens and lower quality of life. In 2017, Medicare spent nearly $80,000 per person per year for patients with ESRD in the United States, amounting to $35.9 billion in total spending for end-stage renal disease patients [6].

The increasing number of patients with diabetic kidney disease underscores inadequacies in current management approaches, including a lack of aggressive interventions and understanding of the interventions that are most efficacious [7]. Guidelines suggest renoprotective drugs, such as angiotensin-converting enzyme (ACE) inhibitors and angiotensin II receptor blockers (ARBs), should be prescribed to decrease elevated blood pressure in type 2 diabetes mellitus patients at high risk of kidney decline, as these drugs are beneficial in preventing further kidney impairment [8–11]. Guidelines also suggest sodium/glucose cotransporter-2 (SGLT2) inhibitors should be prescribed to decrease the risk of diabetic kidney disease progression in type 2 diabetes mellitus patients [9, 11, 12]. SGLT2 inhibitors benefit patients at risk of kidney decline by reducing the risk of dialysis, transplantation, and even death due to diabetic kidney disease in type 2 diabetes mellitus patients [4]. Despite evidence supporting the benefits of ACE inhibitors, ARBs, and SGLT2 inhibitors, providers' prescription patterns for patients with diabetic kidney disease are not aligned with guideline standards, and current prescription rates of these renoprotective drugs are low in type 2 diabetes mellitus patients who could benefit from these medications [8, 9, 11–14]. In addition, studies found that in adults with diabetic kidney disease, potentially nephrotoxic agents such as nonsteroidal anti-inflammatory drugs (NSAIDs) and proton-pump inhibitors (PPIs) are prescribed frequently despite the potential harm posed to high-risk diabetic kidney disease patients by the long-term utilization of these agents [8, 9, 11, 12].

Traditional measures of kidney function are insufficient for identifying patients at risk of diabetic kidney disease or rapid renal decline. Guidelines from the American Diabetes Association (ADA) and the National Kidney Foundation Kidney Disease Outcomes Quality Initiative (KDOQI) recommend type 2 diabetes mellitus patients receive annual measurement of estimated glomerular filtration rate (eGFR) and annual screening for albuminuria, which is identified by calculating the urinary albumin to creatinine ratio (ACR) (9,12). Diabetic kidney disease is diagnosed by reduced eGFR rate (eGFR $< 60$ mL/min/1.73 m$^2$) and/or elevated albuminuria ($> 30$ mg/g creatinine) that persists at least 3 months [9, 12]. However, albuminuria and eGFR detect diabetic kidney disease only after kidney damage has already occurred and cannot accurately predict the risk of developing diabetic kidney disease prior to kidney function decline [15]. With the incorporation of biomarkers, however, the inadequacies of current methods of diagnosing and monitoring diabetic kidney disease can be mitigated [16–21]. Early identification of patients at risk of diabetic kidney disease or rapid renal decline may slow or prevent diabetic kidney disease progression to ESRD by indicating appropriate interventions and monitoring frequency, as well as by encouraging patient referral to nephrologists when necessary [22, 23]. Proper management can also decrease the economic burden of diabetic kidney disease and reduce the use of unnecessary and costly treatments [11, 22–24].

PromarkerD is a simple biomarker-based blood test that can predict the risk of diabetic kidney disease in adults with type 2 diabetes mellitus up to four years before clinically significant renal decline occurs, as well as predict the risk of rapid renal decline in patients with diabetic kidney disease [25]. The test incorporates three novel protein biomarkers: apolipoprotein A-IV (ApoA4), CD5 antigen-like (CD5L) and insulin-like growth factor-binding protein (IGFBP3/IBP3), in addition to three clinical factors measured at time of the test: age, high-density lipoprotein (HDL) cholesterol and eGFR [26]. These data are submitted to the PromarkerD Hub, a software tool that contains an algorithm that generates a low-, moderate-, or high-risk result and provides an interpretation of the risk score. In validated clinical studies, PromarkerD identified 86% of people with diabetes without renal impairment at baseline who went on to develop chronic kidney disease within four years [25]. PromarkerD also has a negative predictive value, or "rule-out" capability, of 98% for a four-year risk of developing diabetic kidney disease [25]. This study evaluates the impact of PromarkerD versus other type 2 diabetes mellitus patient attributes in informing patient management decisions of primary care physicians (PCPs) and endocrinologists.

## Methods

### Subject selection

For this study, a web-based survey was administered to a convenience sample of PCPs and endocrinologists via an external recruiting organization with proprietary panels of verified medical providers. Respondents were allowed to participate in the study if they (1) were board certified in family medicine, primary care, internal medicine, diabetology, or endocrinology, (2) had at least two years of experience in managing patients, (3) managed over 20 type 2 diabetes mellitus patients in the last six months, (4) tested over 10 type 2 diabetes mellitus patients for diabetic kidney disease in the past six months, and (5) spent at least 50% of their time in direct clinical care. PCPs completing the survey were offered $20 as an honorarium. Initially, endocrinologists were offered $30 as an honorarium; the honorarium was increased to $50 to recruit the remaining 146 endocrinologists needed to meet the study's quota.

### Conjoint analysis experimental design

A conjoint analysis methodology was used to design, implement, and analyze the web-based survey following International Society for Pharmacoeconomics and Outcomes Research

**Table 1. Detailed attributes and levels.**

| Attribute | Level 1 | Level 2 | Level 3 | Level 4 |
|---|---|---|---|---|
| PromarkerD result | No test | Low risk | Moderate risk | High risk |
| Albuminuria | 15 mcg/mg (mildly increased) | 165 mcg/mg (moderately increased) | 500 mcg/mg (severely increased) | Not applicable |
| eGFR | 110 ml/min/1.73m$^2$ (normal) | 75 ml/min/1.73m$^2$ (mildly decreased) | 45 ml/min/1.73m$^2$ (moderately decreased) | Not applicable |
| Blood pressure | 120/70 mmHg | 135/90 mmHg | 150/95 mmHg | Not applicable |
| Glycemic control (HbA1c) | 6.3% | 7.5% | 8.4% | Not applicable |
| Age | 48 years | 66 years | 83 years | Not applicable |

eGFR: estimated glomerular filtration rate; HbA1c: hemoglobin A1c.

(ISPOR) report guidelines for best research practices for a conjoint analysis [27–29]. Conjoint analysis is a method of using standardized hypothetical vignettes, in this case patient profiles, to derive the importance of attributes (e.g., patient characteristics or test results) based on respondents' selection of hypothetical outcomes (e.g., treatment decisions) [27, 29]. This method provides a way to simulate respondent choices in the real world, in which physicians assess multiple patient characteristics collectively before making treatment decisions. Thus, conjoint analysis is a valuable tool for quantifying the implicit trade-off decisions physicians make when assessing management options for patients with type 2 diabetes mellitus [30–32].

Attributes that impact physician decision-making for type 2 diabetes mellitus patients were defined based on a literature review and discussion with two diabetic kidney disease clinical experts (MS and AT) [33–43]. In addition to the PromarkerD result, the attributes were albuminuria, eGFR, blood pressure, glycemic control represented by hemoglobin A1c (HbA1c) level, and age. The PromarkerD result attribute had four possible levels, while the five other attributes were assigned three possible levels. Two levels of eGFR (normal and mildly decreased), in conjunction with a mildly increased albuminuria level, were intended to represent patients without diabetic kidney disease. Other eGFR/albuminuria combinations were selected to characterize patients with diabetic kidney disease. Detailed levels for each attribute are shown in Table 1.

Hypothetical patient profiles were developed using one level from each attribute. A total of 972 unique profiles were possible given the number of attributes and levels (full factorial design). To minimize respondent burden, a subset of profiles was created using Sawtooth Software's Conjoint Value Analysis (CVA) Module version 3.0. This design of 42 unique patient profiles had a D-efficiency of 0.97, indicating that the subset is representative of the full factorial design because it is close to the maximum possible D-efficiency of 1 for a full factorial design [44]. The design was orthogonal, meaning each pair of levels (across different attributes) is intended to appear an equal number of times [27]. To minimize standard errors (<0.1) for a conjoint exercise designed for 400 respondents and to reduce respondent fatigue, the survey software randomly selected eight patient profiles from the subset of 42 profiles. The eight profiles were selected using a least fill methodology, which allowed each of the 42 profiles to be shown approximately the same number of times.

Treatment decision outcomes for the conjoint analysis were chosen based on expert (MS and AT) feedback on key patient management decisions for type 2 diabetes mellitus patients. To provide context and control for confounding variables, the survey prefaced the treatment decision questions by instructing physicians to assume all patients currently: (1) have standard monitoring frequency, (2) have been receiving 10 mg of lisinopril per day, (3) have been receiving 800 mg of ibuprofen per day (i.e., 400 mg twice a day for back pain), (4) have no contraindications to or higher-than-average risk of side-effects from SGLT2 inhibitors or ACE

inhibitors, and (5) have been receiving the maximum tolerated dose of metformin. S1 Fig provides a full list of patient assumptions. For each patient profile presented, respondents were then asked to select "yes" or "no" for the following three questions: (1) "Would you prescribe SGLT2s that have a diabetic kidney disease indication?"; (2) "Would you increase lisinopril dose to 20 mg per day for renoprotection?"; and (3) "Would you replace ibuprofen with a non-systemic therapy (e.g., a lidocaine patch or topical NSAIDs) or continue ibuprofen?". Respondents were also asked to select "increase," "decrease," or "maintain standard monitoring frequency" for the fourth question: "What would be your recommendation for monitoring risk factors (albuminuria, eGFR, blood pressure, and HbA1c)?" S2 Fig shows an example of a conjoint analysis patient profile and treatment decision questions.

## Survey development

The self-administered web survey was designed to take 15 minutes to complete. To ensure data quality, survey responses were reviewed and physicians who took fewer than five minutes to finish the study were excluded from the final data set. The survey comprised four sections. The first section obtained background information on type 2 diabetes mellitus patients' typical care. In the second section, respondents were shown a blinded description of PromarkerD (described throughout the survey as "Test X"), including information on the data collection methods, test result interpretations, and test validation data. This section evaluated respondents' likelihood to adopt PromarkerD in the absence of cost considerations, assuming the test was commercially available, FDA-approved, and adopted by the respondent's practice. The third section included the conjoint experiment and evaluated how management and prescribing patterns would vary based on attribute levels in the patient profiles. The fourth section assessed respondents' reactions to PromarkerD. Materials presented to respondents (PromarkerD profile and assumptions) are included in (S1–S3 Figs). S3 Fig presents the accuracy data that existed at the time the survey was in the field: sensitivity (86%), specificity (78%), and negative predictive value (98%) for incident DKD. Similar test performance characteristics for a combined group of patients at risk for either incident DKD or progression of DKD are now available: the sensitivity, specificity, and negative predictive value associated with a combined incident DKD or eGFR30 result are 87%, 83%, and 97%, respectively [45].

## Survey deployment

Five respondents (3 endocrinologists and 2 PCPs) took the survey as a pilot test to identify any elements of the study requiring further clarification or modification. To evaluate test-retest consistency, the pilot version of the survey supplemented the typical eight patient profiles with two additional identical patient profiles (hold-out cases) not included in the original set of 42 patient profiles. Based on the pilot interviews, the text describing the attribute levels was refined before final launch with all respondents. For example, ranges for each level were replaced with point estimates and interpretive clinical descriptions of point values were added for eGFR levels (i.e., normal, mildly, or moderately decreased) and albuminuria levels (i.e., mildly, moderately, or severely increased). Sterling Institutional Review Board (IRB), an individual board, reviewed and determined this study to be exempt from a full IRB review. Following the pilot tests, the survey was open to physicians from October 27, 2020 to November 9, 2020.

## Data analysis

Data analysis was conducted with Sawtooth Software (Menu-based Choice: MBC v1.1 2016). In the analysis, the categories "decrease monitoring frequency" and "maintain standard

monitoring frequency" of the monitoring frequency outcome variable were combined due to sparse data, as "decrease monitoring frequency" was selected as an option for only 5% of the patient profiles. Consequently, this monitoring outcome was analyzed as a binary variable (increase in monitoring frequency versus no increase in monitoring frequency). A multivariable logit model for each outcome (prescription of SGLT2 inhibitors, increase in lisinopril dose, replacement of ibuprofen with pain therapies that did not involve systemic NSAIDs, and increase in monitoring frequency) was used to analyze the impact of PromarkerD and other patient attributes on physician decision-making. Each of the four models produced relative utilities for each attribute level. For each model, the relative importance of each attribute in physician decision-making was calculated as the difference between the minimum and maximum utility for that attribute; the relative importance values were then normalized to sum to 100%. Additionally, utilities were used to generate odds ratios, with 95% confidence intervals, to evaluate the decision impact of PromarkerD results versus no test; odds ratios were also calculated to compare other patient attribute levels with a reference level selected for each attribute. Separate models for each specialty (PCPs and endocrinologists) were also assessed to identify any significant differences between specialties, but specialty-specific values derived from the conjoint analysis are not included in this manuscript given the increased standard error associated with the smaller sample size for each specialty.

## Results

### Demographic characteristics

A total of 7,851 web survey invitations were emailed to prospective participants; 596 physicians started the survey, and 405 completed it, including five physicians who pilot tested the survey and are not included in the final analysis. Thus, the final sample included 400 physicians, the predetermined target number of physicians for this research. The median time to finish the survey was 15 minutes. Fig 1 describes the sample attrition leading to the final number of respondents in this study.

Table 2 describes characteristics of the 400 qualified respondents who completed the survey following the pilot tests. Forty-nine percent of respondents noted endocrinology or diabetology as their primary specialty, and 51% listed their primary specialty as primary care/internal medicine/family medicine. A majority (66%) of respondents indicated that their primary practice was an office-based private practice. Eighteen percent of respondents practice predominantly in community hospitals and 16% in academic medical centers. The geographic location of respondents was relatively evenly distributed. The geographic distribution of the survey participants was representative of the geographic distribution in the United States [46].

### Descriptive findings

For type 2 diabetes mellitus patients at low risk of diabetic kidney disease, current monitoring of diabetic kidney disease risk factors (blood pressure, HbA1c, albuminuria levels, and eGFR levels) ranges from 87% of physicians for eGFR (172/197 endocrinologists and 175/203 PCPs) to 93% of physicians for blood pressure (81/197 endocrinologists and 191/203 PCPs). Figs 2 and 3 present physician responses regarding risk factor monitoring.

All physicians reported monitoring one or more risk factors in type 2 diabetes mellitus patients at low risk of diabetic kidney disease at least once per year. For type 2 diabetes mellitus patients at high risk of developing diabetic kidney disease, the percentage of physicians monitoring risk factors at least every six months was 99% for physicians who monitor HbA1c (195/197 endocrinologists and 201/203 PCPs), 94% for physicians who monitor eGFR (180/194 endocrinologists and 192/201 PCPs), 74% for physicians who monitor albuminuria (145/197

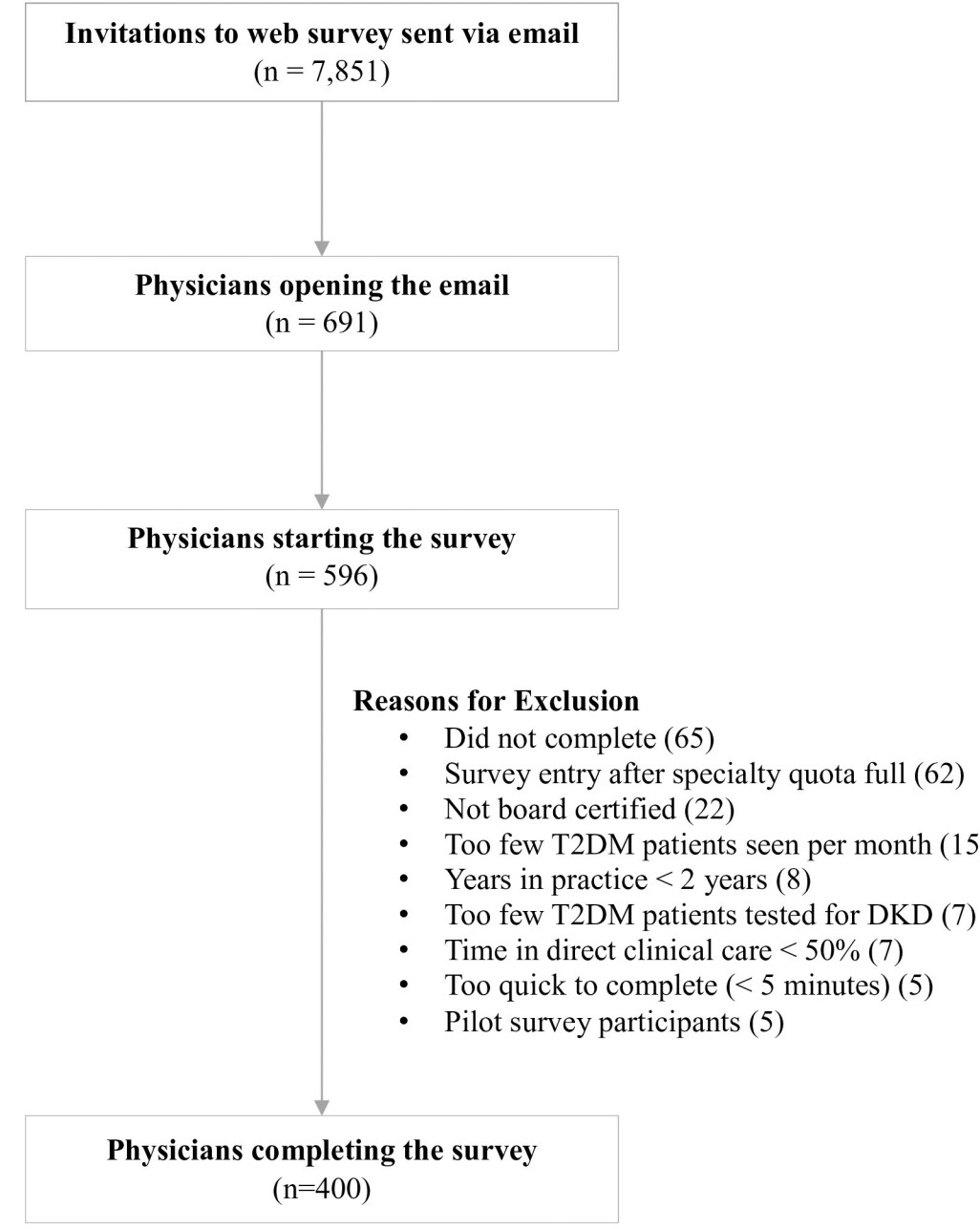

DKD: diabetic kidney disease; T2DM: type-2 diabetes mellitus.

**Fig 1. Attrition of survey respondents.**

endocrinologists and 151/202 PCPs), and 98% for physicians who monitor blood pressure (193/ 197 endocrinologists and 197/203 PCPs).

Sixty-two percent of physicians (134/197 endocrinologists and 113/203 PCPs) reported they prescribe SGLT2 inhibitors in type 2 diabetes mellitus patients at low risk of diabetic kidney disease. An additional 18% of physicians (24/197 endocrinologists and 48/203 PCPs) indicated that while they would not prescribe SGLT2 inhibitors in type 2 diabetes mellitus patients

**Table 2. Subject characteristics table.**

| Provider/ Practice Variables | Respondents (n = 400) | Percentage |
|---|---|---|
| Primary Specialty | | |
| Endocrinology (Including Diabetology) | 197 | 49% |
| Primary Care/ Internal Medicine/ Family Medicine | 203 | 51% |
| Setting of Care | | |
| Office-Based Private Practice | 264 | 66% |
| Community Hospital | 72 | 18% |
| Academic Medical Center | 64 | 16% |
| Geographic Distribution | | |
| Midwest | 76 | 19% |
| Northeast | 72 | 18% |
| South | 144 | 36% |
| West | 108 | 27% |

at low risk of diabetic kidney disease, they would order SGLT2 inhibitors for their patients at an increased risk of diabetic kidney disease. To manage blood pressure, 90% of physicians surveyed (162/197 endocrinologists and 196/203 PCPs) currently prescribe ACE inhibitors/ARBs in type 2 diabetes mellitus patients at low risk for diabetic kidney disease. Of those physicians

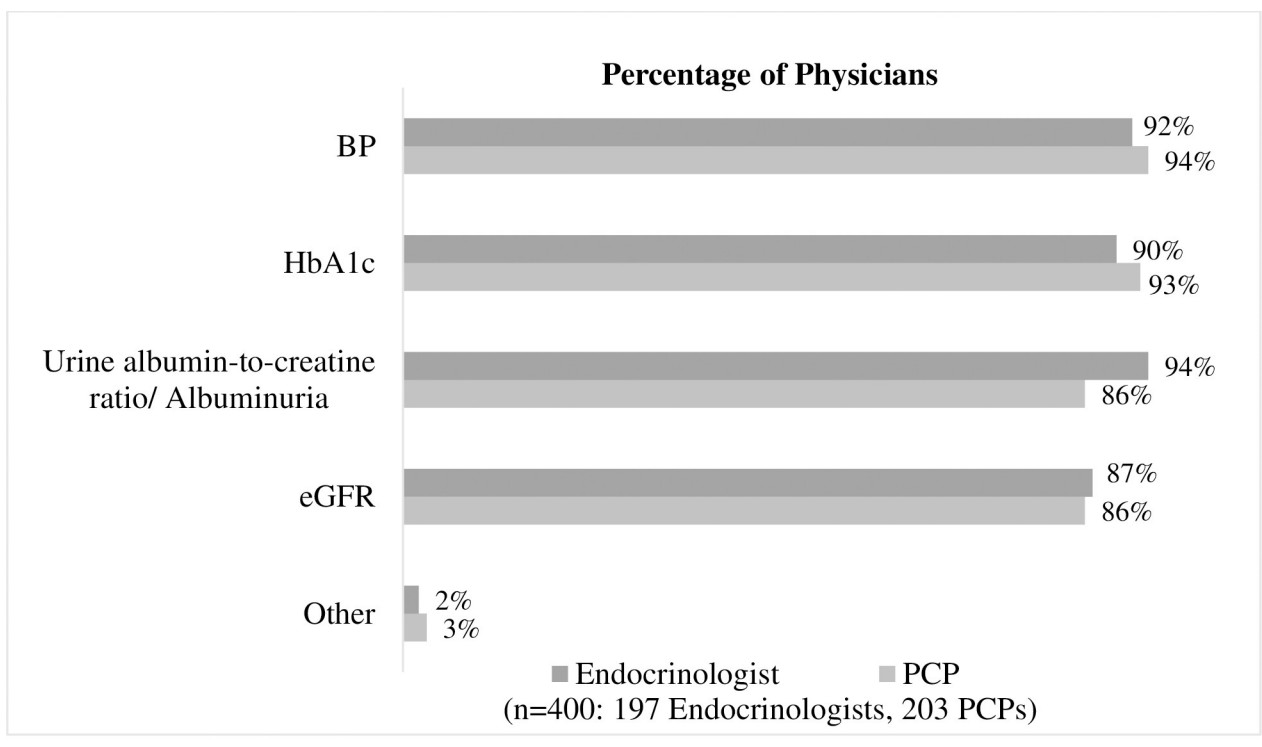

[a]More than one response accepted.

BP: blood pressure; eGFR: estimated glomerular filtration rate; HbA1c: hemoglobin A1c;

PCP: primary care physician.

**Fig 2. Tests conducted in type 2 diabetes mellitus patients at low risk of developing diabetic kidney disease.**

**A. Patients at *low risk* for diabetic kidney disease**

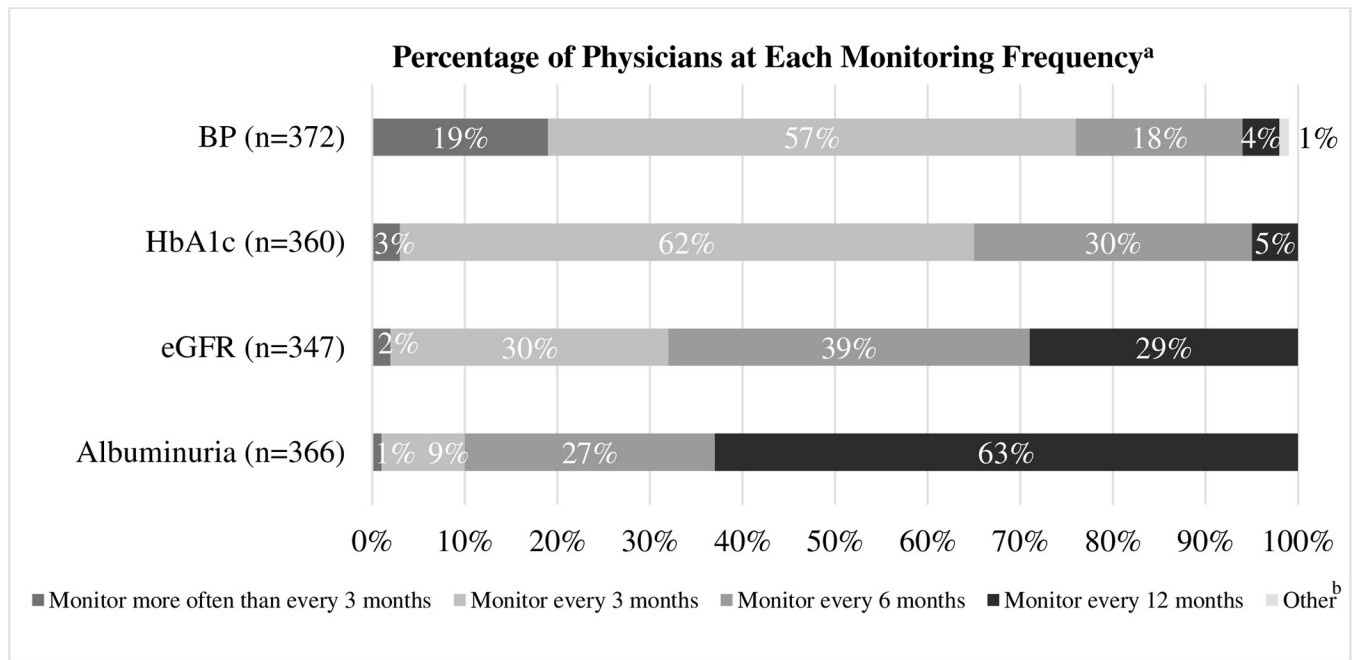

**B. Patients at *increased risk* for diabetic kidney disease**

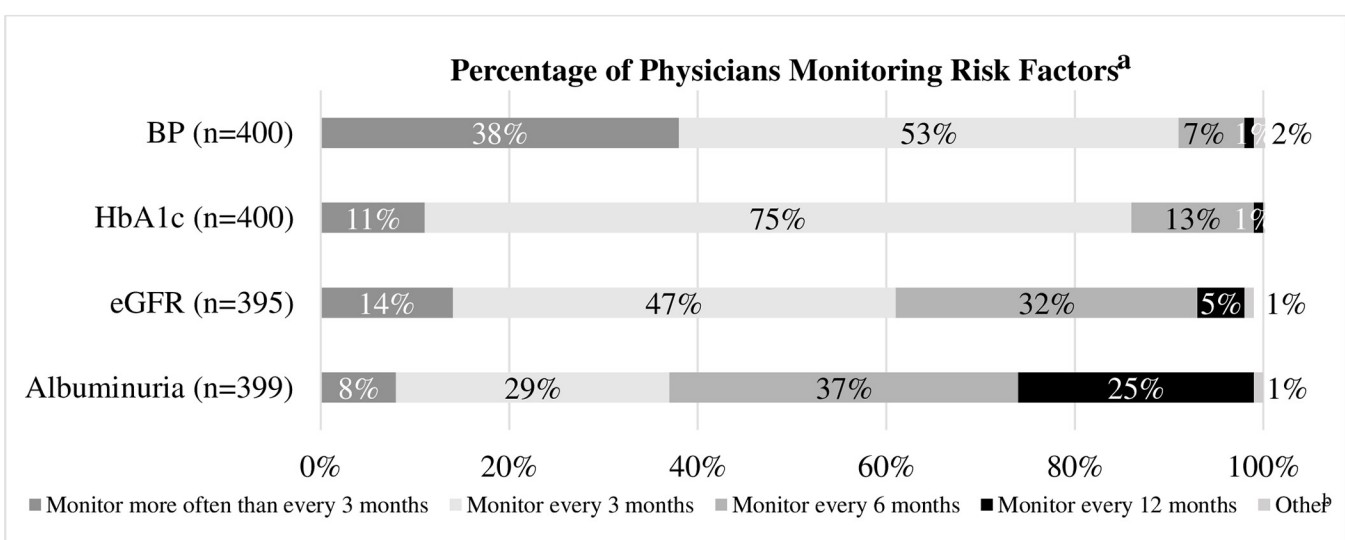

[a]Percentages listed for each attribute may not add to 100% as a result of rounding.

[b]A small percentage of physicians responded "other," which includes daily, monthly, every two months, every four months, and every office visit.

BP: blood pressure; eGFR: estimated glomerular filtration rate; HbA1c: hemoglobin A1c; PCP: primary care physician.

**Fig 3. Frequency of monitoring of type 2 diabetes mellitus patients.**

who currently prescribe ACE inhibitors/ARBs to low-risk patients, 94% (157/162 endocrinologists and 179/196 PCPs) noted they would increase the dose of ACE inhibitors/ARBs in type 2 diabetes mellitus patients at an increased risk of diabetic kidney disease.

Sixty-three percent of endocrinologists (124/197) and 67% of PCPs (136/203) acknowledged the difficulty of using current tools to predict the progression of diabetic kidney disease, and 44% of endocrinologists (87/197) and 46% of PCPs (93/203) indicated predicting the onset of diabetic kidney disease in the near future for type 2 diabetes mellitus patients is a challenge.

Seventy-eight percent of physicians indicated they are very or extremely likely to order PromarkerD in their type 2 diabetes mellitus patients following FDA approval of the test. Physicians expected to order PromarkerD in 66% of type 2 diabetes mellitus patients, on average. Additionally, 84% of physicians (154/197 endocrinologists and 174/203 PCPs) would consider PromarkerD results when deciding whether to refer a patient to a nephrologist, and 58% (102/197 endocrinologists and 130/203 PCPs) indicated PromarkerD would serve as motivation for patients to make lifestyle changes. Only 3% of endocrinologists (5/197) and 1% of PCPs (3/203) indicated they would not order PromarkerD in any of their patients, regardless of the cost.

## Conjoint analysis findings

For each patient management decision, Fig 4 presents the relative importance of patient attributes based on the range of utility scores in the aggregate logit models. For the decision to increase monitoring, a PromarkerD result was the most important attribute (relative importance: 35.8%), followed by albuminuria levels (19.7%). For the decision to prescribe SGLT2 inhibitors, HbA1c level was the most important attribute (39.2%) followed by PromarkerD result (25.5%). For the decision to increase the dose of lisinopril, blood pressure was the most important attribute (53.0%), followed by the PromarkerD result (12.2%). For replacing ibuprofen with a non-nephrotoxic pain medication, eGFR was the most important attribute (31.0%), followed by a PromarkerD result (28.4%).

Odds ratio (OR) values, calculated for patient attribute levels from the utility scores of the aggregate logit models, are presented in (S1–S4 Tables); all significance determinations are based on α <0.05. For each patient management decision, Fig 5 summarizes OR values with 95% confidence intervals (CIs) specifically for PromarkerD results compared with no test, while controlling for other attributes. Compared with not having the test available, PromarkerD moderate- and high-risk test results were associated with significantly greater odds of increasing monitoring frequency (OR: 1.57, 95% CI: 1.15–2.13 and OR: 2.56, 95% CI: 1.90–3.45, respectively), prescribing SGLT2 inhibitors (OR: 1.50, 95% CI: 1.19–1.90 and OR: 1.98, 95% CI: 1.56–2.52, respectively), and replacing ibuprofen (OR: 1.60, 95% CI: 1.18–2.18 and OR: 1.78, 95% CI: 1.32–2.40, respectively). A high-risk PromarkerD result was associated with significantly greater odds of increasing the dose of lisinopril (OR: 1.48, 95% CI: 1.17–1.87). Fig 5 also shows that a low-risk PromarkerD score resulted in significantly lower odds of increasing monitoring frequency (OR: 0.48, 95% CI: 0.37–0.64), prescribing SGLT2 inhibitors (OR: 0.70, 95% CI: 0.56–0.88), and replacing ibuprofen (OR: 0.75, 95% CI: 0.57–0.99) compared with no test.

## Discussion

This study aimed to determine the influence of PromarkerD on physician-decision making in type 2 diabetes mellitus patients and the importance of the test compared with the current standard of care testing to assess risk of diabetic kidney disease and renal decline. Physician

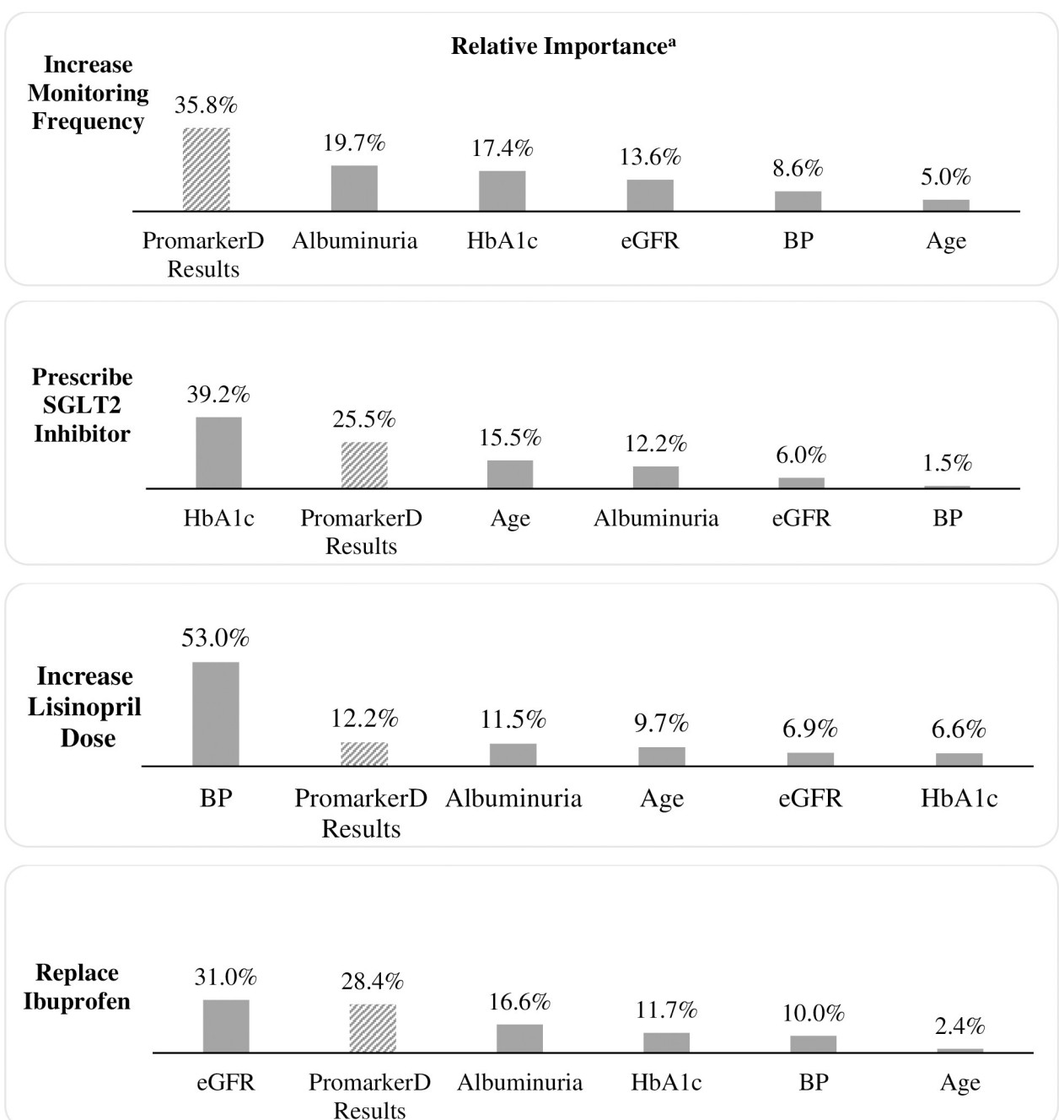

<sup>a</sup>While attribute importance values sum to 100% across attributes for each outcome assessed, percentages listed may not add to 100% as a result of rounding.

BP: blood pressure; eGFR: estimated glomerular filtration rate; HbA1c: hemoglobin A1c.

**Fig 4. The relative importance of each attribute in influencing measured outcomes.**

data from this investigation demonstrated that endocrinologists and PCPs ascribed a clinical benefit to predicting the risk of diabetic kidney disease in type 2 diabetes mellitus patients before kidney damage occurs, as well as predicting rapid renal decline in patients with diabetic

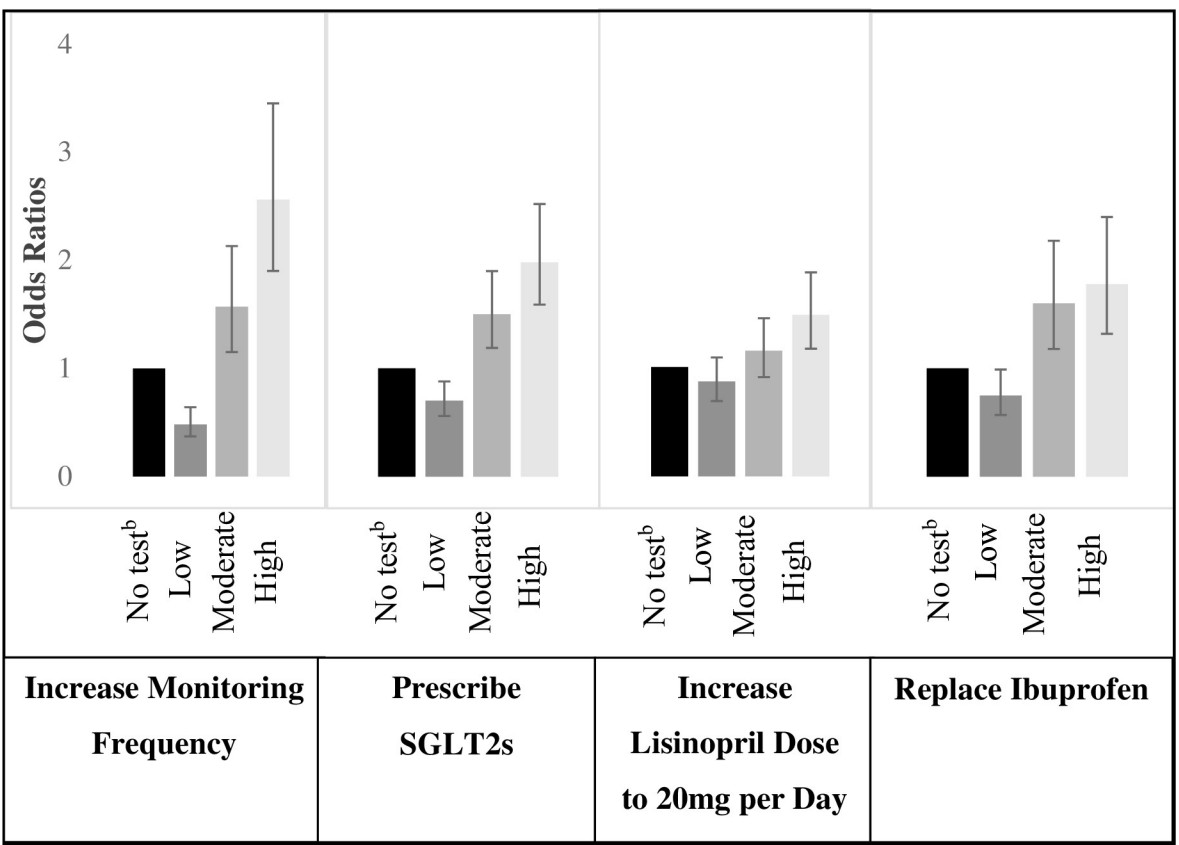

[a]Vertical lines represent the 95% confidence interval for the odds ratio estimates.

[b]"No test" was the reference level.

SGLT2: sodium/glucose cotransporter-2

**Fig 5. Impact of PromarkerD results on physician decision-making.**

kidney disease; with accurate risk prediction, physicians would be able to administer optimal interventions and treatments at the appropriate time, understand when to prioritize the avoidance of nephrotoxic drugs, and tailor risk factor monitoring frequency to each patient.

This study indicated that the PromarkerD result was the most important or second most important attribute leading to a change in physician behavior across all modeled clinical decisions. PromarkerD was the second most important attribute, ahead of albuminuria and eGFR, leading to the increase in dose of lisinopril (an ACE inhibitor for blood pressure) and the prescription of SGLT2s. In addition, PromarkerD was the second most important attribute contributing to the replacement of ibuprofen with non-nephrotoxic pain medications, ahead of albuminuria levels. The conjoint exercise also indicated PromarkerD was more important for physician decisions to increase risk factor monitoring frequency than all other modeled attributes, including both albuminuria and eGFR levels. Collectively, these findings show that routine clinical factors (albuminuria and eGFR) were less influential in decision-making for predicting the onset of diabetic kidney disease and rapid renal decline in type 2 diabetes mellitus patients compared with a novel test such as PromarkerD.

The findings align with studies indicating the potential value of combining biomarkers with more traditional patient metrics to assess the risk of diabetic kidney disease [16–21]. Data from this study are also consistent with research on the inadequacy of tools to predict diabetic kidney disease and the opportunity for improved management of type 2 diabetes mellitus patients at risk of diabetic kidney disease or renal function decline. Colhoun and Marcovecchio (2018) concluded that current kidney function tests are insufficient for predicting the onset or progression of diabetic kidney disease in type 2 diabetes mellitus patients [47]. Braun et al. (2012) underscored the complex nature of current standard of care methods to determine optimal patient care strategies for type 2 diabetes mellitus patients and the need to improve methods of identifying patients at high risk for diabetic kidney disease [15].

This study indicated that physicians not only would value the information provided by PromarkerD but would also apply the test results appropriately to maximize the opportunity to improve patients' health. A low-risk PromarkerD result would significantly reduce the likelihood of aggressive patient management compared with no test, suggesting PromarkerD may reduce unnecessary treatments, limiting adverse effects and saving time and costs. Similarly, a PromarkerD moderate- and high-risk result would significantly increase the likelihood of renoprotective treatment changes by physicians and would support a more personalized approach to diabetic kidney disease management. The importance of using elevated PromarkerD risk results to identify patients at risk of diabetic kidney disease before renal damage occurs could increase even further in the future as new medications are developed for kidney disease in type 2 diabetes mellitus patients.

The primary strength of this study was the study design. The conjoint analysis allowed the importance of PromarkerD and other attributes to be inferred implicitly from physician decision-making instead of requiring physicians to comment on the importance of these factors directly, which can bias responses [29]. The conjoint study also provided a way to simulate physician behavior in the real world, in which physicians assess multiple patient characteristics conjointly when managing patients.

The study has several limitations. First, the findings are limited to the levels of the attributes included in this analysis and may not completely reflect the impact of every patient characteristic that could influence physicians' treatment and monitoring decisions. However, the attributes and levels included in this study were ascertained from secondary research and feedback from physicians who specialize in type 2 diabetes mellitus and diabetic kidney disease (MS and AT). The survey was also tested with five physicians to confirm appropriate design and attribute levels. Thus, the attributes and levels represented the most important characteristics for decision-making. Second, while the study was intended to represent real world decision-making, survey results may still differ from actual physician behavior. However, the discrete-choice study methodology is an objective methodology validated and endorsed by ISPOR [27]. Variations of this study design have been utilized for studies assessing patient preferences in diabetes care, as well as for studies evaluating factors that affect testing decisions of primary care physician in other indications [29, 31]. Third, a small subset of the 7,851 physicians who were invited to participate in the survey (9%) opened the recruiting email, and 596 physicians started the survey. The possibility exists that the final data from the 400 physicians included in the final analysis may represent a biased sample. Nevertheless, the survey data were reviewed to ensure that the included physicians' geographic distribution was similar to the distribution of physicians in the US [46] and that the physicians' practice types represented a range of care settings (community hospital, academic medical center, and office-based practices). Lastly, the study did not assess type 2 diabetes mellitus patient outcomes following physician decisions. Additional research will need to be conducted to assess these outcomes.

## Conclusion

This study suggests PromarkerD would significantly impact physicians' prescribing and monitoring decisions for type 2 diabetes mellitus patients. Seventy-eight percent of physicians indicated they were very or extremely likely to order PromarkerD in their type 2 diabetes mellitus patients. PromarkerD results were relatively more important to physicians than current standard-of-care tests, eGFR and albuminuria. PromarkerD has the potential to increase the number of type 2 diabetes mellitus patients receiving appropriate treatment; moderate- and high-risk PromarkerD results were expected to increase the likelihood of renoprotective changes in management of type 2 diabetes mellitus patients at risk of diabetic kidney disease or rapid renal decline compared with no test results, while low-risk results were expected to lower the likelihood of aggressive treatment and health care resource utilization. Physician data from this study indicate PromarkerD could provide clinical utility in the management of diabetic kidney disease in patients with type 2 diabetes mellitus and offer a cost-effective personalized approach to improving patient outcomes by earlier targeted treatment of those patients at highest risk of diabetic kidney disease.

## Supporting information

**S1 Fig. Assumptions for conjoint exercise.**
(TIF)

**S2 Fig. Sample patient profile and decision-making exercise in conjoint analysis.**
(TIF)

**S3 Fig. Test X (PromarkerD) profile.**
(TIF)

**S1 Table. Association of clinical variables and PromarkerD results with increasing monitoring from standard monitoring frequency (n = 400).**
(DOCX)

**S2 Table. Association of clinical variables and PromarkerD results with prescribing SGLT2 inhibitors having a diabetic kidney disease indication (n = 400).**
(DOCX)

**S3 Table. Association of clinical variables and PromarkerD results with increasing lisinopril dose to 20mg for renoprotection (n = 400).**
(DOCX)

**S4 Table. Association of clinical variables and PromarkerD results with replacing ibuprofen with non-systemic therapy (n = 400).**
(DOCX)

**S1 File. Raw data.**
(XLSX)

## Acknowledgments

The authors would like to acknowledge Dr. Richard Lipscombe for reviewing the manuscript.

## Author Contributions

**Conceptualization:** Lauren Fusfeld, Jessica T. Murphy, YooJin Yoon, Li Ying Kam, Kirsten E. Peters, Pearl Lin Tan.

**Data curation:** Lauren Fusfeld, Jessica T. Murphy, YooJin Yoon.

**Formal analysis:** Lauren Fusfeld, Jessica T. Murphy, YooJin Yoon.

**Funding acquisition:** Li Ying Kam, Kirsten E. Peters, Pearl Lin Tan.

**Investigation:** Lauren Fusfeld, Jessica T. Murphy, YooJin Yoon.

**Methodology:** Lauren Fusfeld, Jessica T. Murphy, YooJin Yoon.

**Project administration:** Lauren Fusfeld.

**Software:** Lauren Fusfeld.

**Supervision:** Lauren Fusfeld.

**Validation:** Michael Shanik, Alexander Turchin.

**Visualization:** Lauren Fusfeld, Jessica T. Murphy, YooJin Yoon.

**Writing – original draft:** Lauren Fusfeld, Jessica T. Murphy, YooJin Yoon.

**Writing – review & editing:** Lauren Fusfeld, Jessica T. Murphy, YooJin Yoon, Li Ying Kam, Kirsten E. Peters, Pearl Lin Tan, Michael Shanik, Alexander Turchin.

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
