## [Decision Letter · Decision Letter 0]

28 Feb 2022

PONE-D-21-22693Evaluation of the clinical utility of the PromarkerD in-vitro test in predicting diabetic kidney disease and rapid renal decline through a conjoint analysisPLOS ONE

Dear Dr. Fusfeld,

Thank you for submitting your manuscript to PLOS ONE. After careful consideration, we feel that it has merit but does not fully meet PLOS ONE’s publication criteria as it currently stands. Therefore, we invite you to submit a revised version of the manuscript that addresses the points raised during the review process.

The manuscript has been evaluated by two reviewers, and their comments are available below.

The reviewers have raised a number of concerns that need attention. They request additional discussion of some of the limitations of the study (such as details of the performance of the test for prevalent DKD), and request additional information regarding methodological aspects of the study (such as why sex was not considered as a attribute).

Could you please carefully revise the manuscript to address all comments raised?

We look forward to receiving your revised manuscript.

Kind regards,

Jamie Royle

Associate Editor

PLOS ONE

Journal Requirements:

2. 

PONE-D-21-22693

"LF and JM are employed by BHA, and YY was employed by BHA at the time of the research. Proteomics employs LYK, KP, and PLT. Dr. Shanik has received consulting fees from Proteomics International. Dr. Turchin has received consulting fees from Proteomics International, has equity in Brio Systems, and has received research funding from Astra Zeneca, Edwards, Eli Lilly, Novo Nordisk, Pfizer, and Sanofi. Proteomics International is a beneficiary of patent PCT/AU2011/001212 that relates to biomarkers described in this manuscript."

Reviewers' comments:

Reviewer's Responses to Questions

**Comments to the Author**

1. Is the manuscript technically sound, and do the data support the conclusions?

Reviewer #1: Yes

Reviewer #2: Partly

2. Has the statistical analysis been performed appropriately and rigorously? 

Reviewer #1: Yes

Reviewer #2: Yes

3. Have the authors made all data underlying the findings in their manuscript fully available?

Reviewer #1: Yes

Reviewer #2: No

4. Is the manuscript presented in an intelligible fashion and written in standard English?

Reviewer #1: Yes

Reviewer #2: Yes

5. Review Comments to the Author

Reviewer #1: This study evaluated the impact to physicians of PromarkerD, a biomarker-based blood test predicting the risk of diabetic kidney disease and rapid renal decline. 400 physicians (203 PCPs and 197 endocrinologists) were surveyed, and the investigators found through conjoint analyses the relative importance of ProMarkerD in making treatment decisions (starting SGLT2i, increasing ACEi dose, or discontinuing nephrotoxins). The analyses are straight forward and easy to understand. The limitations are well-delineated.

There is, however, a major limitation. The test description in Supplementary Figure 3 only describe the performance characteristics of PromarkerD for developing DKD (e.g., new incident DKD). However, Level 2 and 3 attributes for albuminuria (165 and 500 mcg/mg) and level 3 attribute for eGFR (45 ml/min) would indicate prevalent DKD, and there the performance that matters is risk of progression. That is not well described in Supplementary Figure 3. (there are no statistics shown with the risk for 30% decline in eGFR). Thus, it appears that the analyses and surveys were largely applied to hypothetical patients outside the intended use of the test? I would suggest separate analyses that stratify those that did not yet have DKD vs. those that already have DKD.

Reviewer #2: In the present manuscript, authors aimed to evaluate how PromarkerD test would impact physicians’ prescribing and monitoring decisions for type 2 diabetes mellitus patients. Although identification of novel non-invasive biomarkers represents a challenge in the field of diabetic kidney disease, however the study presents several limitations.

Did authors evaluated in a cohort of real patients up if decisions made based on PromarkerD reduced the progression of renal damage in the follow in those patients indicated at high risk and had no impact on renal decline in patient estimated at low risk? An important point is the prediction of rapid renal decline in patients with diabetic kidney disease, since DKD includes different kinds of renal damages with different pathogenesis mechanisms and different treatments. In most of the cases only renal biopsy can estimate the real cause of DKD and support treatment decisions. Thus described cases could not really represent real-life.

It is not clear to me how interpret the PromarkerD results and how levels were assigned. They should include a range for each attribute to assign low risk or moderate risk… Moreover, why sex was not considered among attribute?

Number of diabetic kidney disease clinical experts for treatment decision is limited. Where they nephrologist or endocrinologist?

An important limitation of the survey is also the lack of informations regarding data collection methods, test result interpretations, test validation and costs since a correct evaluation on the decision impact of PromarkerD results versus no test should also consider time and costs needed to obtain data.

6. PLOS authors have the option to publish the peer review history of their article (what does this mean?). If published, this will include your full peer review and any attached files.

Reviewer #1: No

Reviewer #2: No

---

## [Author Response · Author response to Decision Letter 0]

26 Mar 2022

Review Comments to the Authors

Reviewer #1: This study evaluated the impact to physicians of PromarkerD, a biomarker-based blood test predicting the risk of diabetic kidney disease and rapid renal decline. 400 physicians (203 PCPs and 197 endocrinologists) were surveyed, and the investigators found through conjoint analyses the relative importance of ProMarkerD in making treatment decisions (starting SGLT2i, increasing ACEi dose, or discontinuing nephrotoxins). The analyses are straight forward and easy to understand. The limitations are well-delineated.

There is, however, a major limitation. The test description in Supplementary Figure 3 only describe the performance characteristics of PromarkerD for developing DKD (e.g., new incident DKD). 

However, Level 2 and 3 attributes for albuminuria (165 and 500 mcg/mg) and level 3 attribute for eGFR (45 ml/min) would indicate prevalent DKD, and there the performance that matters is risk of progression. That is not well described in Supplementary Figure 3. (there are no statistics shown with the risk for 30% decline in eGFR). Thus, it appears that the analyses and surveys were largely applied to hypothetical patients outside the intended use of the test? 

Author response: Thank you for your thoughtful review and comments.

The test is intended to be used both with patients who are at risk of DKD and patients with DKD who are at risk of further renal decline. At the time the survey was in the field, we did not have the performance characteristics of PromarkerD specifically for disease progression with patients who already have DKD. Since we wanted to present validated accuracy data as opposed to hypothetical accuracy data, we provided the sensitivity, specificity, and NPV numbers associated with the risk of developing DKD, while noting in Supplementary Figure 3, “If the patient’s eGFR level at the time of the test is already <60mL/min/1.73m2, then the risk of further kidney decline in the next four years, defined as an eGFR decline of >30%, is provided.” Prior to launching the survey, two diabetic kidney disease clinical experts (MS and AT) were instrumental in reviewing the profile of PromarkerD for clarity and completeness for the intended audience of endocrinologists and primary care physicians. Later, during video calls with Boston Healthcare, five physicians pilot tested the survey without expressing confusion about the profile of PromarkerD or concern that the data provided in the profile impeded their ability to provide an educated assessment of their medical decisions for each hypothetical patient. In addition, none of the respondents who completed the survey indicated in the open-ended text questions that the test description was inadequate. 

Since the time the survey was fielded, additional performance data have become available. The sensitivity (87%), specificity (83%), and negative predictive value (97%) associated with a combined incident DKD or eGFR30 result are similar to the values for incident DKD alone: sensitivity (86%), specificity (78%), and negative predictive value (98%). We cannot change Supplementary Figure 3 because this figure contains the information the physician respondents actually viewed, but we can add the additional accuracy data. These additional values are now on page 10, lines 179-184.

I would suggest separate analyses that stratify those that did not yet have DKD vs. those that already have DKD.

Author response: While it could be interesting to design and field additional conjoint studies that test the impact of PromarkerD and other attributes with patients at risk of DKD separately from patients who already have DKD, we cannot conduct this analysis post-hoc with the data available. As noted on page 8, lines 137-144, the conjoint design of 42 unique patient profiles was originally created because the design had a D-efficiency of 0.97 (indicating that the subset was representative of the full factorial design) based on all the attributes and levels (including eGFR) and because it was possible to minimize standard errors with the given sample size of 400 respondents. 

Reviewer #2: In the present manuscript, authors aimed to evaluate how PromarkerD test would impact physicians’ prescribing and monitoring decisions for type 2 diabetes mellitus patients. Although identification of novel non-invasive biomarkers represents a challenge in the field of diabetic kidney disease, however the study presents several limitations.

Did authors evaluated in a cohort of real patients up if decisions made based on PromarkerD reduced the progression of renal damage in the follow in those patients indicated at high risk and had no impact on renal decline in patient estimated at low risk? 

Author response: Thank you for your thoughtful review and comments.

We agree that a clinical study like the one you propose using real patients could be helpful in the future. On page 19, lines 366-368, the manuscript acknowledges, “the study did not assess type 2 diabetes mellitus patient outcomes following physician decisions. Additional research will need to be conducted to assess these outcomes.” This research could take the form of a clinical trial such as the one you had described.

The downside of using a clinical study for the initial research, however, is that such a study requires significant resources and costs, particularly during this COVID era. The clinical utility study described in this manuscript was intended to assess whether physicians will use the test in their decision making; the study was not intended to be an assessment the impact of the test on patient outcomes using data from a chart review. 

A conjoint approach is an ISPOR-approved way to understand physicians’ motivation; this validated methodology, using clinical vignettes, was developed to understand implicit decision making and to mimic the real world, as noted on page 18, lines 363-364, (Bridges 2011). This conjoint/discrete choice approach has been proven to be a pragmatic, cost-effective, and time-efficient methodology for firms developing new medical technology. PLOS ONE has published these types of studies in a variety of disease areas (Boone 2013, de Vries 2015, Harrison 2019, Hazlewood 2020; Kromer 2015, and Ride 2016), as have other respected journals (Janssen 2018, Szeinback 2008), noted on page 19, lines 364-366. 

An important point is the prediction of rapid renal decline in patients with diabetic kidney disease, since DKD includes different kinds of renal damages with different pathogenesis mechanisms and different treatments. In most of the cases only renal biopsy can estimate the real cause of DKD and support treatment decisions. Thus described cases could not really represent real-life.

Author response: The literature indicates that physicians are making treatment decisions based on KDIGO and other guidelines and that there is a need for a test like this (Datar 2021). Very few patients with suspected DKD have kidney biopsies in real-life practice of medicine (Looker 2018). In the personal experience of both diabetic kidney disease clinical experts (MS and AT), kidney biopsies are only considered by nephrologists when etiology other than DKD is suspected.

These two diabetic kidney disease clinical experts (MS and AT) were involved in developing the attributes and levels to ensure that the patient profiles were realistic and sufficiently informative for physicians to estimate their medical decisions for each hypothetical patient profile. Five physicians pilot tested the survey without indicating that the data provided in the patient profiles were insufficient for them to provide an informed medical decision. In addition, none of the respondents who completed the survey indicated in the open-ended text questions that the profiles were inadequate. 

It is not clear to me how interpret the PromarkerD results and how levels were assigned. 

Author response: PromarkerD measures 3 plasma biomarkers (ApoA4, CD5L, IGFBP3) combined with 3 clinical factors (age, HDL-cholesterol, eGFR). These data are submitted to the PromarkerD Hub, a software tool that contains an algorithm that generates a low-, moderate-, or high-risk result and provides an interpretation of the risk score. This information regarding how PromarkerD assigns levels can be seen on page 6, lines 86-91. Further details regarding the test are outside the scope of the current manuscript but have been published elsewhere (Bringans 2017; Peters 2017; Peters 2019).

Supplementary Figure 3 provides an explanation of how to interpret the test results. For example, a low-risk result indicates the 4-year risk of developing DKD (or further renal decline for patients with DKD) ranges from 0% to <10%. Similarly, a moderate-risk result indicates the 4-year risk of developing DKD (or further renal decline for patients with DKD) ranges from 10% to <20%, while a high-risk result from PromarkerD indicates the 4-year risk of developing DKD (or further renal decline for patients with DKD) ranges from 20% to 100%. 

They should include a range for each attribute to assign low risk or moderate risk… 

Author response: As noted on pages 10, lines 190-192, although the original text describing the attribute levels included ranges for each level, pilot interviews indicated ranges were too difficult to evaluate, so we replaced the ranges with point estimates.

Moreover, why sex was not considered among attribute?

Author response: We conducted a targeted literature review to identify the patient attributes with the largest impact on physician decision-making for type 2 diabetes mellitus patients; we then narrowed the list of attributes based on discussions with two diabetic kidney disease clinical experts (MS and AT). Including too many attributes would unduly complicate the conjoint choice exercise and would require each physician to review a burdensome number of profiles and/or necessitate an increase in the sample size. 

Sex, in particular, did not qualify for the list of attributes submitted to the clinical experts for review. The impact of sex in the risk of DKD and the risk of DKD progression is unclear. Maric-Bilkan (2020) notes, “Particularly, women with diabetes have higher mortality rates for diabetes-related deaths, and higher prevalence of diabetic kidney disease risk factors such as hypertension, hyperglycemia, obesity, and dyslipidemia. However, the evidence for the impact of sex on diabetic kidney disease prevalence and disease progression is limited and inconsistent. Although most studies agree that the protective effect of the female sex against the development of kidney disease is diminished in the setting of diabetes, the reasons for this observation are unclear. Whether or not sex differences exist in the risk of diabetic kidney disease is also unclear, with studies reporting either higher risk in men, women, or no sex differences.” 

Number of diabetic kidney disease clinical experts for treatment decision is limited. Where they nephrologist or endocrinologist?

Author response: Both clinical experts involved in developing the primary research materials for this study are endocrinologists. We also spoke with five physicians (three endocrinologists and two primary care physicians) in the pilot phase of this study. As noted in Table 2 on pages 12 and 13, 400 additional physicians (197 endocrinologists and 203 primary care physicians) are included in the analysis. 

PromarkerD can help primary care physicians and endocrinologists, the target clinicians in this study, decide whether to refer patients to nephrologists.

An important limitation of the survey is also the lack of informations regarding data collection methods, test result interpretations, test validation and costs since a correct evaluation on the decision impact of PromarkerD results versus no test should also consider time and costs needed to obtain data.

Author response: For data collection, in a real-life clinical setting, the process is as follows:

Clinical Process 

1. A patient with type 2 diabetes mellitus would be seen regularly by the managing clinician (most commonly a general practitioner) for general monitoring and annual standard of care tests, including eGFR and uACR, to assess kidney function.

2. The managing clinician would request PromarkerD for patients with a recent history of eGFR and HDL-cholesterol results. 

3. Patients would be referred for a blood draw.

4. Blood samples would be sent to accredited laboratories where the PromarkerD kit would be used to measure each protein biomarker in the sample via enzyme-linked immunoassay (ELISA).

5. Test results would be interpreted and uploaded into the PromarkerD hub to produce a risk score.

6. The PromarkerD risk score would be provided to managing clinicians who would then relay that information to patients.

7. The test risk score would inform further patient care such as monitoring frequency, lifestyle modification, addition of other medications, initiation of more aggressive treatment measures, and patient education on risk factors.

Supplementary Figure 3 presents test result interpretation guidelines for physician, which are based on well-established KDIGO guidelines.

We agree that in practice clinicians consider many factors, including time and cost, when making patient management decisions. However, we would like to emphasize that this study was intended to be a clinical survey involving hypothetical patients to assess potential use of PromarkerD and to determine where physicians saw clinical value. Assumptions for the choice exercise in survey are provided in Supplementary Figure 1, including patient description, test cost (“The cost of Test X is not an issue for you and your patients”), and test availability. Further details on test validation are outside the scope of the current manuscript but have been published elsewhere (Bringans 2017; Peters 2017; Peters 2019). PromarkerD’s price in the United States has not been determined; a full cost-effectiveness model is currently underway. 

Additional Publications Referenced in this Letter but Not Included in the Manuscript

Conjoint Studies in Medical Research

• Boone D, Mallett S, Zhu S, et al. Patients’ & healthcare professionals’ values regarding true- & false-positive diagnosis when colorectal cancer screening by CT colonography: discrete choice experiment. PLOS ONE. 2013;8(12):e80767.

• de Vries ST, de Vries FM, Dekker T, et al. The role of patients’ age on their preferences for choosing additional blood pressure-lowering drugs: a discrete choice experiment in patients with diabetes. PLOS ONE. 2015;10(10):e0139755.

• Harrison M, Spooner L, Bansback N, et al. Preventing rheumatoid arthritis: Preferences for and predicted uptake of preventive treatments among high risk individuals. PLOS ONE. 2019;14(4):e0216075.

• Hazlewood GS, Pokharel G, Deardon R, et al. Patient preferences for maintenance therapy in Crohn’s disease: A discrete-choice experiment. PLOS ONE. 2020;15(1):e0227635.

• Kromer C, Schaarschmidt ML, Schmieder A, Herr R, Goerdt S, Peitsch WK. Patient preferences for treatment of psoriasis with biologicals: a discrete choice experiment. PLOS ONE. 2015;10(6):e0129120.

• Ride J, Lancsar E. Women’s preferences for treatment of perinatal depression and anxiety: a discrete choice experiment. PLOS ONE. 2016;11(6):e0156629.

Study Demonstrating Need for PromarkerD

• Datar M, Ramakrishnan S, Montgomery E, Coca SG, Vassalotti JA, Goss T. A qualitative study documenting unmet needs in the management of diabetic kidney disease (DKD) in the primary care setting. BMC Public Health. 2021;21(1):930.

Use of Kidney Biopsies

• Looker HC, Mauer M, Nelson RG. Role of kidney biopsies for biomarker discovery in diabetic kidney disease. Adv Chronic Kidney Dis. 2018;25(2):192-201.

PromarkerD Validity

• Bringans SD, Ito J, Stoll T, et al. Comprehensive mass spectrometry based biomarker discovery and validation platform as applied to diabetic kidney disease. EuPA Open Proteom. 2017;14:1-10.

Sex Differences in Kidney Disease

• Maric-Bilkan C. Sex differences in diabetic kidney disease. Mayo Clin Proc. 2020;95(3):587-599.

---

## [Decision Letter · Decision Letter 1]

19 May 2022

PONE-D-21-22693R1Evaluation of the clinical utility of the PromarkerD in-vitro test in predicting diabetic kidney disease and rapid renal decline through a conjoint analysisPLOS ONE

Dear Dr. Fusfeld,

Thank you for submitting your manuscript to PLOS ONE. After careful consideration, we feel that it has merit but does not fully meet PLOS ONE’s publication criteria as it currently stands. Therefore, we invite you to submit a revised version of the manuscript that addresses the points raised during the review process.

We look forward to receiving your revised manuscript.

Kind regards,

Donovan Anthony McGrowder, PhD., MA., MSc

Academic Editor

PLOS ONE

Journal Requirements:

Editor Comments: 

Dear Dr. Fusfeld,

Your manuscript “Evaluation of the clinical utility of the PromarkerD in-vitro test in predicting diabetic kidney disease and rapid renal decline through a conjoint analysis” has been assessed by our reviewers. They have raised a number of points which we believe would improve the manuscript and may allow a revised version to be published in PLOS ONE.

If you are able to fully address these points, we would encourage you to submit a revised manuscript to PLOS ONE.

Reviewers' comments:

Reviewer's Responses to Questions

**Comments to the Author**

1. If the authors have adequately addressed your comments raised in a previous round of review and you feel that this manuscript is now acceptable for publication, you may indicate that here to bypass the “Comments to the Author” section, enter your conflict of interest statement in the “Confidential to Editor” section, and submit your "Accept" recommendation.

Reviewer #3: (No Response)

Reviewer #4: All comments have been addressed

Reviewer #5: (No Response)

Reviewer #6: (No Response)

Reviewer #7: (No Response)

Reviewer #8: (No Response)

Reviewer #9: (No Response)

Reviewer #10: (No Response)

Reviewer #11: All comments have been addressed

Reviewer #12: All comments have been addressed

2. Is the manuscript technically sound, and do the data support the conclusions?

Reviewer #3: Yes

Reviewer #4: Yes

Reviewer #5: Yes

Reviewer #6: (No Response)

Reviewer #7: No

Reviewer #8: Partly

Reviewer #9: Partly

Reviewer #10: Yes

Reviewer #11: Yes

Reviewer #12: (No Response)

3. Has the statistical analysis been performed appropriately and rigorously? 

Reviewer #3: Yes

Reviewer #4: Yes

Reviewer #5: Yes

Reviewer #6: (No Response)

Reviewer #7: Yes

Reviewer #8: Yes

Reviewer #9: Yes

Reviewer #10: I Don't Know

Reviewer #11: Yes

Reviewer #12: (No Response)

4. Have the authors made all data underlying the findings in their manuscript fully available?

Reviewer #3: Yes

Reviewer #4: Yes

Reviewer #5: Yes

Reviewer #6: (No Response)

Reviewer #7: Yes

Reviewer #8: Yes

Reviewer #9: Yes

Reviewer #10: Yes

Reviewer #11: Yes

Reviewer #12: (No Response)

5. Is the manuscript presented in an intelligible fashion and written in standard English?

Reviewer #3: Yes

Reviewer #4: Yes

Reviewer #5: Yes

Reviewer #6: (No Response)

Reviewer #7: Yes

Reviewer #8: Yes

Reviewer #9: Yes

Reviewer #10: (No Response)

Reviewer #11: Yes

Reviewer #12: (No Response)

6. Review Comments to the Author

Reviewer #3: In this manuscript, the authors assessed the impact of PromakerD in clinical decision-making by conjoint analysis in diabetes mellitus patients at risk of worsening renal failure. The reviewer thinks the methodology is novel and the conclusion is well supported by the results. The authors’ response to the previous reviewers looks appropriate.

The reviewers believe that this study is of clinical importance, but would like to identify some issues that have not been raised by previous reviewers.

The authors generated 42 hypothetical patient profiles for the conjoint analysis. Although the authors showed the attributes and levels of the generated patients in Table 1, the reviewer would suggest showing the summarized data of the actual parameters of the hypothesized patients (e.g. the median age with the 95% C.I.).

PromarkerD takes age and eGFR into account, and these two items overlap among the six attributes. Is this not a confounding factor in performing a conjoint analysis? That is, would clinicians who focus on Age and eGFR tend to focus on PromarkerD?

The “Descriptive Findings” contents are difficult to grasp on first reading. Among the supplemental figures (S4-6), the reviewer would suggest that S5 and S6 be combined into one figure and upgraded to the main figure to help readers understand the contents.

Reviewer #4: Fusfeld et al. evaluated the clinical utility of the PromarkerD in 400 physicians for predicting the risk of diabetic kidney disease and rapid renal decline. PromarkerD can successfully predict patients who develop chronic kidney disease (CKD) from previous studies. However, in Type 2 Diabetes in the Canagliflozin Cardiovascular Assessment Study (CANVAS), PromarkerD can predict incident CKD but had limited utility for predicting eGFR decline ≥30% (J Clin Med. 2020 Oct 6;9(10):3212. doi: 10.3390/jcm9103212.). Therefore, it is not well documented for the prediction or rapidly renal function decline. The interpretation of this part should be cautious. Besides, what is the recommendation for additional treatment strategy when patients had a moderate or higher risk PromarkerD result. A clinical implication could also be emphazied in the discussion section.

Reviewer #5: Another major limitation is that only 400 physicians out of approximately 8,000 physicians who were invited eventually joined the study. Please mention the bias in the Discussion section.

Reviewer #6: (No Response)

Reviewer #7: Comments

1. This study attempted to assess the impact of changes in physician practice patterns based on PromarkerD, a study that is becoming well established. However, since PromarkarD is anonymized with Test T, the usefulness of PromarkarD is not disclosed. This is hardly a conclusion that can be drawn from PromarkarD's previous research findings, and participants would have imagined a perfect test with no deficiencies.

2. The responses examined in this study were probably influenced more by basic knowledge, such as whether or not ibuprofen is known to have an effect on renal dysfunction, than by changes in responses due to PromarkarD (TEST X). In order to evaluate the usefulness of PromarkarD, it would be necessary to evaluate the effectiveness of the model without PromarkarD values (association between various parameters and practice patterns for the No TEST cases) and the model with added PromarkarD values for changes in practice patterns.

Reviewer #8: This article describes the utility of PromarkerD, a biomarker-based blood test, in clinical decision making for monitoring renal function in diabetic patients. They used conjoint analysis, a method that derives attribute importance based on respondents' selection of hypothetical outcomes using standardized virtual vignettes (p. 7). After analyzing a web-based survey of decisions of 400 physicians to treat eight from forty-two hypothetical patient profiles, they found that PromarkerD was most or second most important in the decision of treatment strategies including increasing the frequency of risk factor monitoring, prescribing SGLT2s inhibitors with a diabetic kidney disease indication, increasing the dose of lisinopril, and replacing ibuprofen with a non-nephrotoxic medication. They concluded that PromarkerD could increase adoption of renoprotective interventions in patients at high risk and lower the likelihood of aggressive treatment in those at low risk for renal decline.

This study is well designed and clearly explained. The response to reviewers is acceptable.

The reviewer has some minor comments as follows.

In the abstract, it is concluded that PromarkerD could increase adoption of renoprotective interventions in patients at high risk and lower the likelihood of aggressive treatment in those at low risk for renal decline. However, as discussed in the revision, this virtual study could not assess the patient outcomes following physician decisions. Therefore, after the conclusion, the need for further clinical studies to assess patient outcomes after applying this Promarker D in real-world practice should be described in the abstract. This will clarify the limitations of the study design and help readers understand the current position of PromarkerD in clinical application.

References should be added to support the statement on page 10, lines 178-184.

Reviewer #9: I read with great interest the manuscript entitled " Evaluation of the clinical utility of the PromarkerD in-vitro test in predicting diabetic kidney disease and rapid renal decline through a conjoint analysis".

This study surveyed and assessed the interventional behavior of primary care physicians and nephrologists following the results of PromarkerD, a test that was approved to predict the progression of diabetic nephropathy. They found that this marker is taken into consideration as a secondary parameter after HbA1c, blood pressure levels and eGFR to act upon prescribing SGLT2i, lisinopril or avoiding ibuprofen respectively.

This is a well-written paper. The research question is interesting and can be the first step to evaluate in the future whether the decisions of physicians based on this new marker would improve the progression of CKD in patients with diabetes.

I have two comments.

1. Minor comment to be taken into consideration or not by authors and editor (Reference: https://kdigo.org/wp-content/uploads/2022/03/KDIGO-2022-Diabetes-Management-GL_Public-Review-draft_1Mar2022.pdf):

Regarding the nomenclature of kidney diseases, there is a tendency, adopted by the KDIGO and that will be launched soon, suggesting to avoid the term "diabetic kidney disease" because we are never sure without a biopsy that diabetes is the main cause of chronic kidney disease. So KDIGO team will call to rather use "chronic kidney disease with diabetes". The KDIGO replaced as well the term "end-stage kidney disease" by "kidney failure". However, a lot of authors worldwide are still using the terms used in this manuscript.

2. The sample of surveyed physicians was a convenience sample. The survey was sent to 7851 physicians and 405 completed it. The authors did not mention if the sample size was representative of physicians taking care of chronic kidney disease patients with diabetes. Who is the target population in this study, is it the American pool of physicians treating diabetic kidney disease? Or is it only the 7851 selected physicians? Even if it is only the 7851 physicians and we assume a confidence level of 95%, a margin of error of 4% and the fact that 50% of the physicians treat CKD patients with diabetes, the sample size should be 558 physicians. The authors need to discuss this in the limitations of the study.

Another concern related to this issue: inside the methodology, the authors state that they designed the survey and profiles based on the 400 respondents. Does it mean that they were aiming to stop the survey at 400 respondents? This needs a clarification.

Reviewer #10: To evaluate the impact to physicians of PromarkerD, a biomarker-based blood test predicting the risk of diabetic kidney disease and rapid renal decline, the authors performed a web-based survey asked each physician to make monitoring and treatment decisions about eight randomly selected profiles using a conjoint analysis in 400 physicians. The results indicated that PromarkerD result was most important for increasing the frequency of risk factor monitoring, and second in importance for deciding to prescribe sodium/glucose cotransporter-2 inhibitors (SGLT2s) with a diabetic kidney disease indication, increasing the dose of lisinopril, and replacing ibuprofen with a non-nephrotoxic medication. A high-risk PromarkerD result was associated with significantly higher odds of increasing monitoring frequency, prescribing SGLT2, increasing lisinopril dose, and replacing ibuprofen, whereas a low-risk PromarkerD result was associated with significantly lower odds of increasing monitoring frequency, prescribing SGLT2s, and replacing ibuprofen compared with no PromarkerD results. According to these findings, the authors concluded that PromarkerD could increase adoption of renoprotective interventions in patients at high risk and lower the likelihood of aggressive treatment in those at low risk for renal decline.

The theme of this study is interesting and the manuscript is well written.

I have some comments.

1. How much did the subjects know about PromarkerD? I am asking this because I believe that the subjects’ knowledge of PromarkerD would reflect the results of this study. What are the authors’ thoughts on this point?

2. It would be interesting to stratify this analysis by primary care physicians and endocrinologists and examine the difference between the two groups. The author described “Separate models for each specialty (PCPs and endocrinologists) were also assessed to identify any significant differences between specialties, but specialty-specific values derived from the conjoint analysis are not included in this manuscript given the increased standard error associated with the smaller sample size for each specialty” (lines 215-218), but it seems possible because the sample size is not small, with approximately 200 subjects in each group.

3. In accordance with the report of KDIGO Consensus Conference regarding nomenclature for kidney function and disease (Levey AS, et al. Kidney Int 97: 1117-1129, 2020), I propose to change the term “renal” to “kidney”; that is, “rapid renal decline” and “end-stage renal disease” should be changed to “rapid kidney decline” (or “rapid decline in kidney function” would be better) and “end-stage kidney disease”, respectively.

4. The word “ESRD” (line 78) should be spelled out.

Reviewer #11: This revised paper was well-written and well responded for reviewer's comments. Promarker D would be useful to prevent DKD progression. We hope this useful marker could reduce the cost for dialysis for ESRD by diabetes.

Reviewer #12: (No Response)

---

## [Author Response · Author response to Decision Letter 1]

30 Jun 2022

Reviewer #3: In this manuscript, the authors assessed the impact of PromarkerD in clinical decision-making by conjoint analysis in diabetes mellitus patients at risk of worsening renal failure. The reviewer thinks the methodology is novel and the conclusion is well supported by the results. The authors’ response to the previous reviewers looks appropriate.

The reviewers believe that this study is of clinical importance, but would like to identify some issues that have not been raised by previous reviewers.

The authors generated 42 hypothetical patient profiles for the conjoint analysis. Although the authors showed the attributes and levels of the generated patients in Table 1, the reviewer would suggest showing the summarized data of the actual parameters of the hypothesized patients (e.g. the median age with the 95% C.I.).

Author response: Thank you for your thoughtful review and suggestions. 

Sawtooth Software’s CVA module was used to generate a design that is orthogonal (meaning that each pair of levels across different attributes appears about an equal number of times) and balanced (meaning within each attribute, each level appears about an equal number of times). The table below shows the number of times each level of each attribute appears in the data set of this study. Based on the attribute level counts (i.e., the number of times the attribute level appears in a profile) in this table, the different age levels are almost equally represented in the conjoint design. 

Attribute Level Counts

 Attributes

Attribute Levels Att 1 - Albuminuria (ACR) (mg/g) Att 2 - eGFR Att 3 - Blood Pressure Att 4 - HbA1c Att 5 - Age Att 6 - Test result

1 993 1063 1068 990 1066 835

2 1143 992 1064 1065 1067 762

3 1064 1145 1068 1145 1067 768

4 NA NA NA NA NA 835

The full data set for the conjoint analysis has been provided as part of the submission, and the manuscript text on page 8, line 143 notes the orthogonality. We believe that describing the design in this manner is more appropriate for this type of analysis than providing median values. 

PromarkerD takes age and eGFR into account, and these two items overlap among the six attributes. Is this not a confounding factor in performing a conjoint analysis? That is, would clinicians who focus on Age and eGFR tend to focus on PromarkerD?

The goal of the conjoint exercise was to understand physicians’ decision-making in an exercise designed to mimic real-world situations in which doctors have access to PromarkerD test results and other patient characteristics, including age and eGFR. While it is possible that clinicians who focus on age and eGFR might also focus on PromarkerD (which incorporates these factors), it is also possible that clinicians could disregard the results of PromarkerD and focus solely on currently available patient attributes. The conjoint analysis answers the question, “Are age and eGFR individually more valuable than PromarkerD, which includes them as part of a composite score?” The data indicate that the importance of PromarkerD is actually much higher than the importance of age and eGFR individually. 

In addition, while we were designing the conjoint exercise, we confirmed that every level of a given attribute is realistically able to occur with every level of other attributes. Hence, no prohibitions exist between the PromarkerD results and the age levels presented (i.e., the low, moderate, and high PromarkerD results could occur with any of the ages in the conjoint analysis); likewise, no prohibitions exist between the PromarkerD results and the eGFR results presented. 

The “Descriptive Findings” contents are difficult to grasp on first reading. Among the supplemental figures (S4-6), the reviewer would suggest that S5 and S6 be combined into one figure and upgraded to the main figure to help readers understand the contents.

Thank you for your suggestion. Because the original Supplemental figures S4-6 provide background information about the current state of care (tests conducted in type 2 diabetes patients and monitoring frequency in patients at low and increased risk for diabetic kidney disease) rather than information about the conjoint exercise itself, we had originally placed the figures in the Supplemental section. However, we do want to make the findings as easy to understand as possible, so we have moved them into the main document at your recommendation. Supplemental figure S4 is now Figure 2 and is mentioned on page 13, lines 248-249. We also appreciate that it may be useful to compare the results of S5 and S6, so we have combined the figures into a single figure in the main text (mentioned on Figure 3, page 14, line 250) as you have suggested.

Reviewer #4: Fusfeld et al. evaluated the clinical utility of the PromarkerD in 400 physicians for predicting the risk of diabetic kidney disease and rapid renal decline. PromarkerD can successfully predict patients who develop chronic kidney disease (CKD) from previous studies. However, in Type 2 Diabetes in the Canagliflozin Cardiovascular Assessment Study (CANVAS), PromarkerD can predict incident CKD but had limited utility for predicting eGFR decline ≥30% (J Clin Med. 2020 Oct 6;9(10):3212. doi: 10.3390/jcm9103212.). Therefore, it is not well documented for the prediction or rapidly renal function decline. The interpretation of this part should be cautious. 

Author response: Thank you for your methodical review and recommendations. 

We acknowledge that the performance of the eGFR decline ≥30% component of PromarkerD was poorer in CANVAS than in previous studies, but a discussion of this was beyond the scope of the present study. Nevertheless, one possible cause of the performance difference is the variation in the distribution of PromarkerD scores with respect to cut-off values across studies. In CANVAS, the distribution of PromarkerD scores for the eGFR decline ≥30% outcome was highly skewed, with most participants (>98%) having low-risk scores and none having high-risk scores. Given this skewed distribution, there was limited power to detect even small effect sizes for PromarkerD moderate-risk scores versus low-risk scores on outcome. Nevertheless, higher PromarkerD scores were predictive of outcome, when analyzed as a continuous score from 0-100% (OR 1.13 95% CI 1.04-1.24), indicating that PromarkerD was indeed able to predict eGFR decline ≥30%, and that the risk cut-offs may need adjusting in some populations.

There are also several demographic and clinical differences between CANVAS and FDS2. PromarkerD was developed in a relatively healthy population of people with well-controlled type 2 diabetes. Subjects in CANVAS had higher glycated hemoglobin and either had a history of cardiovascular disease or were at high-risk of future cardiovascular events based on risk factors (long diabetes duration, hypertension, current smoking, albuminuria, or low serum HDL-cholesterol). In contrast, FDS2 is a community-based cohort that is representative of patients with T2D from an urban Australian population. Compared with FDS2, CANVAS participants at baseline were younger (mean age 62.7 versus 65.5 years), more were male (67.0% versus 54.3%), the median glycated hemoglobin was higher (8.0% versus 6.8%), and the median diabetes duration was longer (12.4 versus 7.0 years). In terms of baseline renal function, CANVAS participants had more renal impairment (16.5% with eGFR <60 mL/min/1.73m2) but less albuminuria (27.8% with uACR ≥30 mg/g) compared with those in FDS2 (11.6% with renal impairment and 34.2% with albuminuria).

Besides, what is the recommendation for additional treatment strategy when patients had a moderate or higher risk PromarkerD result. 

Supplementary figure 3 (mentioned on page 10, line 181) describes the treatment strategy after moderate-risk results and high-risk results. The treatment strategy is based on the consensus report from the American Diabetes Association and the European Association for the Study of Diabetes (Davies MJ et al. Management of hyperglycemia in type 2 diabetes, 2018. A consensus report by the American Diabetes Association (ADA) and the European Association for the Study of Diabetes (EASD). Diabetes Care. 2018;(41):2669–2701.)

The treatment strategy for patients with a moderate-risk result includes the following: more frequent risk factor monitoring (every 3-6 months); optimization of lifestyle factors (nutrition, weight loss, exercise, and quitting smoking/drinking); a review of glycemic targets/management; review of non-glycemic risk factors/management, including blood pressure (e.g., ACEi, ARBs) and lipids (e.g., statins); avoidance of potentially nephrotoxic drugs (e.g., NSAIDs); and consideration of the use of drugs with evidence of renoprotection (e.g., SGLT2i, GLP1ra).

The treatment strategy for patients with a high-risk result includes the following: very close risk factor monitoring (every 1-3 months); intensive management strategies based on those for moderate risk, with optimization of treatments for diabetes and other risk factors; and possible referral to a nephrologist for maximal intervention.

A clinical implication could also be emphasized in the discussion section.

We agree the clinical impact of PromarkerD is important. We have provided commentary on the clinical impact in the original submission in the Discussion section on page 17, lines 318-321; the text in this section states, “With accurate risk prediction, physicians would be able to administer optimal interventions and treatments at the appropriate time, understand when to prioritize the avoidance of nephrotoxic drugs, and tailor risk factor monitoring frequency to each patient.” Also, on page 18, lines 347-355, the Discussion section notes, “A low-risk PromarkerD result would significantly reduce the likelihood of aggressive patient management compared with no test, suggesting PromarkerD may reduce unnecessary treatments, limiting adverse effects and saving time and costs. Similarly, a PromarkerD moderate- and high-risk result would significantly increase the likelihood of renoprotective treatment changes by physicians and would support a more personalized approach to diabetic kidney disease management. The importance of using elevated PromarkerD risk results to identify patients at risk of diabetic kidney disease before renal damage occurs could increase even further in the future as new medications are developed for kidney disease in type 2 diabetes mellitus patients.” 

Reviewer #5: Another major limitation is that only 400 physicians out of approximately 8,000 physicians who were invited eventually joined the study. Please mention the bias in the Discussion section.

Author response: Thank you for your recommendation. 

While 7,851 invitations to the survey were sent, 691 opened the email invitation. (We do not know if the unopened invitations were seen by the physicians.) We have added text on the potential for bias in the Discussion section on page 19, lines 374-381.

Reviewer #6: (No Response)

Reviewer #7: Comments

1. This study attempted to assess the impact of changes in physician practice patterns based on PromarkerD, a study that is becoming well established. However, since PromarkerD is anonymized with Test T, the usefulness of PromarkerD is not disclosed. This is hardly a conclusion that can be drawn from PromarkerD's previous research findings, and participants would have imagined a perfect test with no deficiencies.

Author response: Thank you for taking the time to provide comments. 

To ensure that all respondents were basing their responses on the same information about PromarkerD (as opposed to other data or varying conceptions of the test), respondents were presented with a single anonymized description of the test. The test description does include the usefulness of PromarkerD, and the description does not state the test has perfect accuracy. Supplementary figure 3 (mentioned on page 10, line 182) notes, “Unlike existing standards of care, which cannot predict the risk of developing DKD, Test X predicts four-year risk of DKD before clinical symptoms appear. In validated clinical studies, Test X predicted 86% of otherwise healthy diabetics who went on to develop DKD within four years. Test X also predicts the risk of rapid renal decline in DKD patients.” In addition, Supplementary figure 3 notes the sensitivity is 86%, the specificity is 78%, and the negative predictive value is 98%. 

2. The responses examined in this study were probably influenced more by basic knowledge, such as whether or not ibuprofen is known to have an effect on renal dysfunction, than by changes in responses due to PromarkerD (TEST X). In order to evaluate the usefulness of PromarkerD, it would be necessary to evaluate the effectiveness of the model without PromarkerD values (association between various parameters and practice patterns for the No TEST cases) and the model with added PromarkerD values for changes in practice patterns.

The model already accounts for the impact on physician medical decisions of having Test X results (low-risk, moderate-risk, or high-risk) versus not having any Test X results at all. For the decision to replace ibuprofen, for example, the 95% confidence intervals of the odds ratios for low-, moderate-, and high-risk Test X results exclude 1 (the odds ratio representing the reference level of “no test”). The implication is that the test result impacted the decision to replace ibuprofen, as the odds ratios summarize the impact of the test while controlling for patient attributes. Please refer to Figure 5 (mentioned on page 16, line 310), Supplementary table 1D (mentioned on page 16, line 296), and page 16, lines 295-309 of the manuscript for details on the odds ratios.

Reviewer #8: This article describes the utility of PromarkerD, a biomarker-based blood test, in clinical decision making for monitoring renal function in diabetic patients. They used conjoint analysis, a method that derives attribute importance based on respondents' selection of hypothetical outcomes using standardized virtual vignettes (p. 7). After analyzing a web-based survey of decisions of 400 physicians to treat eight from forty-two hypothetical patient profiles, they found that PromarkerD was most or second most important in the decision of treatment strategies including increasing the frequency of risk factor monitoring, prescribing SGLT2s inhibitors with a diabetic kidney disease indication, increasing the dose of lisinopril, and replacing ibuprofen with a non-nephrotoxic medication. They concluded that PromarkerD could increase adoption of renoprotective interventions in patients at high risk and lower the likelihood of aggressive treatment in those at low risk for renal decline.

This study is well designed and clearly explained. The response to reviewers is acceptable.

The reviewer has some minor comments as follows.

In the abstract, it is concluded that PromarkerD could increase adoption of renoprotective interventions in patients at high risk and lower the likelihood of aggressive treatment in those at low risk for renal decline. However, as discussed in the revision, this virtual study could not assess the patient outcomes following physician decisions. Therefore, after the conclusion, the need for further clinical studies to assess patient outcomes after applying this PromarkerD in real-world practice should be described in the abstract. This will clarify the limitations of the study design and help readers understand the current position of PromarkerD in clinical application.

Author response: Thank you for your careful review and suggestions. 

At your recommendation, we have added to the abstract that further clinical studies are needed to assess patient outcomes after applying PromarkerD in real-world practice on page 3, lines 33-35. (Note that to adhere to the word limit of 300 words, we needed to modify some of the other text in the abstract.)

References should be added to support the statement on page 10, lines 178-184.

We agree that the following text requires a reference, “Similar test performance characteristics for a combined group of patients at risk for either incident DKD or progression of DKD are now available: the sensitivity, specificity, and negative predictive value associated with a combined incident DKD or eGFR30 result are 87%, 83%, and 97%, respectively.” We have added a reference on page 10, line 186.

Reviewer #9: I read with great interest the manuscript entitled " Evaluation of the clinical utility of the PromarkerD in-vitro test in predicting diabetic kidney disease and rapid renal decline through a conjoint analysis".

This study surveyed and assessed the interventional behavior of primary care physicians and nephrologists following the results of PromarkerD, a test that was approved to predict the progression of diabetic nephropathy. They found that this marker is taken into consideration as a secondary parameter after HbA1c, blood pressure levels and eGFR to act upon prescribing SGLT2i, lisinopril or avoiding ibuprofen respectively.

This is a well-written paper. The research question is interesting and can be the first step to evaluate in the future whether the decisions of physicians based on this new marker would improve the progression of CKD in patients with diabetes.

I have two comments.

1. Minor comment to be taken into consideration or not by authors and editor (Reference: https://kdigo.org/wp-content/uploads/2022/03/KDIGO-2022-Diabetes-Management-GL_Public-Review-draft_1Mar2022.pdf):

Regarding the nomenclature of kidney diseases, there is a tendency, adopted by the KDIGO and that will be launched soon, suggesting to avoid the term "diabetic kidney disease" because we are never sure without a biopsy that diabetes is the main cause of chronic kidney disease. So KDIGO team will call to rather use "chronic kidney disease with diabetes". The KDIGO replaced as well the term "end-stage kidney disease" by "kidney failure". However, a lot of authors worldwide are still using the terms used in this manuscript.

Author response: Thank you for your thorough review and suggestions. 

We appreciate you calling our attention to the new nomenclature. This manuscript was originally submitted to PLOS ONE in July 2021, before the KDIGO recommendations you cite, and throughout the original survey we use the phrase “diabetic kidney disease.” To avoid confusion, we will refrain from changing the terminology in this manuscript but will keep your recommendation in mind for future research. 

2. The sample of surveyed physicians was a convenience sample. The survey was sent to 7851 physicians and 405 completed it. The authors did not mention if the sample size was representative of physicians taking care of chronic kidney disease patients with diabetes. Who is the target population in this study, is it the American pool of physicians treating diabetic kidney disease? Or is it only the 7851 selected physicians? Even if it is only the 7851 physicians and we assume a confidence level of 95%, a margin of error of 4% and the fact that 50% of the physicians treat CKD patients with diabetes, the sample size should be 558 physicians. The authors need to discuss this in the limitations of the study.

We appreciate this question, as it is an important one to address. If this were a simple regression model at the clinician level, your method of calculating sample size would be accurate. However, this analysis is based on a multi-level model design involving multiple patient profiles that are nested within a single clinician. In other words, this is a patient profile level analysis rather than just a physician level analysis. 

The final multi-level design was determined using dummy data, executing the regression models, and looking at standard errors (<0.1). As noted on page 8, lines 140-148, the conjoint design of 42 unique patient profiles was created because the design had a D-efficiency of 0.97 (indicating that the subset was representative of the full factorial design) based on all the attributes and levels and because it was possible to minimize standard errors with the given sample size of 400 respondents, which was the target number of respondents. In particular, we determined that a minimum of 8 profiles per respondent were necessary to reduce the standard errors (<0.1) for the conjoint design as a whole and for each attribute as recommended by Orme (MBC v1.1: Software for Menu-based choice analysis. 2016. https://sawtoothsoftware.com/uploads/sawtoothsoftware/originals/mbcmanual.pdf).

The low standard errors (<0.1) generated with dummy data from 400 physicians provide strong evidence that our study is adequately powered, as does the fact that the number of profiles generated (3,200) is greater than the sample size of 558 noted above.

According to Bridges et al. (Conjoint Analysis Applications in Health—a Checklist: A Report of the ISPOR Good Research Practices for Conjoint Analysis Task Force. Value Heal. 2011 Jun;14(4):403–13), “Sample-size calculations are particularly difficult for conjoint analysis applications in health care. The appropriate sample size depends on the question format, the complexity of the choice tasks, the desired precision of the results, the degree of heterogeneity in the target population, the availability of respondents, and the need to conduct subgroup analyses. Historically, researchers commonly applied rules of thumb, based on the number of attribute levels, to estimate sample size. Orme recommended sample sizes of at least 300 with a minimum of 200 respondents per group for subgroup analysis. Marshall et al. reported that the mean sample size for conjoint analysis studies in health care published between 2005 and 2008 was 259, with nearly 40% of the sample sizes in the range of 100 to 300 respondents.”

Another concern related to this issue: inside the methodology, the authors state that they designed the survey and profiles based on the 400 respondents. Does it mean that they were aiming to stop the survey at 400 respondents? This needs a clarification.

We have added text on page 12, line 227 to clarify that we had determined that the research would involve 400 respondents prior to starting the research project.

Reviewer #10: To evaluate the impact to physicians of PromarkerD, a biomarker-based blood test predicting the risk of diabetic kidney disease and rapid renal decline, the authors performed a web-based survey asked each physician to make monitoring and treatment decisions about eight randomly selected profiles using a conjoint analysis in 400 physicians. The results indicated that PromarkerD result was most important for increasing the frequency of risk factor monitoring, and second in importance for deciding to prescribe sodium/glucose cotransporter-2 inhibitors (SGLT2s) with a diabetic kidney disease indication, increasing the dose of lisinopril, and replacing ibuprofen with a non-nephrotoxic medication. A high-risk PromarkerD result was associated with significantly higher odds of increasing monitoring frequency, prescribing SGLT2, increasing lisinopril dose, and replacing ibuprofen, whereas a low-risk PromarkerD result was associated with significantly lower odds of increasing monitoring frequency, prescribing SGLT2s, and replacing ibuprofen compared with no PromarkerD results. According to these findings, the authors concluded that PromarkerD could increase adoption of renoprotective interventions in patients at high risk and lower the likelihood of aggressive treatment in those at low risk for renal decline.

The theme of this study is interesting and the manuscript is well written.

I have some comments.

1. How much did the subjects know about PromarkerD? I am asking this because I believe that the subjects’ knowledge of PromarkerD would reflect the results of this study. What are the authors’ thoughts on this point?

Author response: Thank you for your detailed review, questions, and suggestions. 

We do not believe the respondents had much knowledge about PromarkerD (if any) at the time the study was fielded (in 2020). To ensure that all respondents were basing their responses on the same information about PromarkerD (as opposed to other data or varying conceptions of the test), respondents were presented a single anonymized description of the test. Also, during the phone interviews with five physicians during the survey testing phase, the physicians did not identify the test as PromarkerD after reading the description of the test. 

2. It would be interesting to stratify this analysis by primary care physicians and endocrinologists and examine the difference between the two groups. The author described “Separate models for each specialty (PCPs and endocrinologists) were also assessed to identify any significant differences between specialties, but specialty-specific values derived from the conjoint analysis are not included in this manuscript given the increased standard error associated with the smaller sample size for each specialty” (lines 215-218), but it seems possible because the sample size is not small, with approximately 200 subjects in each group. 

Author response: The manuscript is already quite detailed. We feel that adding the additional results by sub-specialty would over-complicate the findings and detract from the main messaging. In addition, some of the other reviewers believe that even 400 respondents represents a small sample size. Consequently, while we agree a stratification analysis would be interesting, we are refraining from including such an analysis in this manuscript.

3. In accordance with the report of KDIGO Consensus Conference regarding nomenclature for kidney function and disease (Levey AS, et al. Kidney Int 97: 1117-1129, 2020), I propose to change the term “renal” to “kidney”; that is, “rapid renal decline” and “end-stage renal disease” should be changed to “rapid kidney decline” (or “rapid decline in kidney function” would be better) and “end-stage kidney disease”, respectively.

We appreciate you calling our attention to the nomenclature. Because the original survey uses the word “renal” in several places, we will refrain from changing the terminology in this manuscript to avoid confusion. Nevertheless, we will keep your recommendation in mind for future research.

4. The word “ESRD” (line 78) should be spelled out. 

You are correct that we should spell out the first mention of ESRD. We have made the change to what is now page 4, line 44.

---

## [Editor Report · Decision Letter 2]

7 Jul 2022

Evaluation of the clinical utility of the PromarkerD in-vitro test in predicting diabetic kidney disease and rapid renal decline through a conjoint analysis

PONE-D-21-22693R2

Dear Dr. Fusfeld,

We’re pleased to inform you that your manuscript has been judged scientifically suitable for publication and will be formally accepted for publication once it meets all outstanding technical requirements.

Kind regards,

Donovan Anthony McGrowder, PhD., MA., MSc

Academic Editor

PLOS ONE

Additional Editor Comments:

Dear Dr. Fusfeld,

The manuscript was revised in accordance with the reviewers’ comments and is provisionally accepted pending final checks for formatting and technical requirements.

Regards,

Dr. Donovan McGrowder (Academic Editor)<o:p></o:p>

---

## [Editor Report · Acceptance letter]

21 Jul 2022

PONE-D-21-22693R2 

Evaluation of the clinical utility of the PromarkerD in-vitro test in predicting diabetic kidney disease and rapid renal decline through a conjoint analysis 

Dear Dr. Fusfeld:

I'm pleased to inform you that your manuscript has been deemed suitable for publication in PLOS ONE. Congratulations! Your manuscript is now with our production department. 

Kind regards, 

on behalf of

Dr. Donovan Anthony McGrowder 

Academic Editor

PLOS ONE